# Big Cooperative Learning to Conquer Local Optima

## Abstract

Cutting-edge foundation models have sparked a groundbreaking AI revolution in a wide range of sophisticated real-world applications. In stark contrast, conventional machine learning paradigms, even with perfect data and model capacity, still persist in grappling with entrenched challenges that manifest in rudimentary forms; for instance, "simple" clustering with mixture models (based on maximum likelihood learning) suffers severely from bad local optima with an exponentially high probability. The marked discrepancy between the achievements of the two research strands gives rise to a question: what is the core element absent from conventional learning paradigms? To answer this question, we assume ideal setup for both data and model capacity and focus on the learning perspective to present the big cooperative learning. Specifically, big cooperative learning makes diverse use of the available (data or energy landscape) information to design massive cooperative training tasks, whose local optima are different but whose global optimum is the same; therefore, by randomly switching among such tasks, big cooperative learning destabilizes and thus conquers their local optima and concurrently encourages exploring the global optimum. Tailored mixture-model-based simulations on forward and reverse KL minimizations (representing the popular maximum likelihood and adversarial learning paradigms, respectively) demonstrate its general effectiveness across multiple paradigms in an explicit and controlled setup.

## 1. Introduction

The world is experiencing a groundbreaking AI revolution with cooperation among scientists and the rise of foundation models (Bommasani et al., 2021; Yuan et al., 2022), such as the GPT series (OpenAI, 2023; 2022; Ouyang et al., 2022; Brown et al., 2020), Sora (Brooks et al., 2024), Geminis (Reid et al., 2024; Team et al., 2023), DALL-Es (Ramesh et al., 2021; 2022; Betker et al., 2024), and BERTs (Devlin et al., 2018; Lan et al., 2019; Liu et al., 2019), which brings advanced AI capabilities to a wide range of sophisticated real-world applications with impressive robustness (Stickland & Murray, 2019; Ramesh et al., 2021; He et al., 2021) and lights up the way towards AI agents (Xi et al., 2023; Guo et al., 2024; Wang et al., 2024a) or even Artificial General Intelligence (AGI) (Fei et al., 2022; Liu et al., 2023; Sun et al., 2024).

However, in stark contrast to the compelling AI power that extensively conquers sophisticated real-world challenges, conventional machine learning paradigms still persist in grappling with entrenched challenges that manifest in rudimentary forms, demonstrating puzzling discrepancy. For example, in the territory of popular maximum likelihood learning (or equivalently forward Kullback-Leibler (KL) minimization), "simple" clustering with mixture models is theoretically proven to suffer from arbitrarily worse local optima than the global optimum with an exponentially high probability (Jin et al., 2016; Chen et al., 2024b). Similarly, "basic" adversarial learning (represented by reverse KL minimization) is notoriously susceptible to mode collapse/seeking, which is also a manifestation of bad local optima (Minka et al., 2005; Srivastava et al., 2017).

The marked discrepancy between the achievements of the two research strands gives rise to the question: if we can overcome sophisticated real-world challenges, we should be able to conquer rudimentary conventional entrenched challenges as well, so what is the core element absent from conventional learning paradigms?

Before answering that question, we first note that, although the great successes of foundation models are often attributed to their large-scale training data and huge model architectures, both data and model capacity are almost certainly *non-ideal*. Conversely, conventional machine learning paradigms, even with *perfect* data and model capacity in controlled simulations, still fail to address their entrenched challenges (as empirically demonstrated in the experiments). Therefore, the aforementioned missing core element must

[1]Anonymous Institution, Anonymous City, Anonymous Region, Anonymous Country. Correspondence to: Anonymous Author <anon.email@domain.com>.

Preliminary work. Under review by the International Conference on Machine Learning (ICML). Do not distribute.

be associated with the learning process.

To answer that question in a clarifying and straightforward manner, we make ideal assumptions on both data (or unnormalized energy landscapes) and model capacity in order to focus on the application-agnostic learning perspective for investigation. Specifically, we observe that the learning of existing foundation models makes diverse use of the available data information, while conventional learning paradigms often monotonously exploit the available (data) information in the *joint* space. Drawing upon these observations, we condense the learning essence of existing foundation models and generalize it into the presented big cooperative learning, which is deemed the missing core element.

Given the available information that could be manifested as data samples from the underlying data distribution or its unnormalized energy landscape, big cooperative learning diversely exploits the available information from versatile perspectives to design massive multi-task training tasks, whose local optima are different but whose global optimum is the same. Accordingly, by followup randomly switching among such tasks, big cooperative learning destabilizes and thus conquers their *inconsistent* local optima and, at the same time, encourages exploring the *consistent* global optimum, demonstrating cooperation among tasks.

To explicitly demonstrate the principle of big cooperative learning, we deliberately bypass situations with black-box Deep Neural Networks (DNNs), where neither local nor global optima are easily accessible. Alternatively, we design tailored simulations based on Gaussian Mixture Models (GMMs), because ($i$) GMMs are representative in that a GMM is a universal approximator of densities (Lindsay, 1995; Peel & MacLahlan, 2000; Goodfellow et al., 2016), ($ii$) GMMs are easy to interpret as their joint, marginal, and conditional distributions are analytic in any linearly transformed domain, and ($iii$) GMMs' local optima are well studied (Chen et al., 2024b). We will briefly discuss situations with DNNs when necessary.

Tailored GMM-based simulations on both forward and reverse KL minimizations (representing the popular maximum likelihood and adversarial learning paradigms, respectively) explicitly demonstrate the general effectiveness of big cooperative learning in conquering local optima across multiple conventional learning paradigms, revealing a promising research direction. Our main contributions include:

- We present the general learning concept of big cooperative learning that is condensed and generalized from the learning of successful foundation models.

- We reveal that big cooperative learning conquers local optima and encourages exploring the global optimum by diversely exploiting the available information (*i.e.,* data samples or unnormalized energy landscapes).

- We design tailored simulations to explicitly demonstrate its general effectiveness in addressing entrenched challenges of multiple conventional paradigms.

## 2. Setup and Preliminaries

### 2.1. Setup

For generalizability, we formulate **the available information** via a Probability Distribution Function (PDF) $q(\boldsymbol{x})$, where $\boldsymbol{x} \in \mathbb{R}^{L \times D}$ has $L$ tokens of dimension $D$ ($D = 1$ unless stated otherwise). Often the available information of the underlying $q(\boldsymbol{x})$ is either manifested as *i.i.d.* data samples $\{\boldsymbol{x}\}$ or its unnormalized energy landscape $\varepsilon(\boldsymbol{x})$ such that $q(\boldsymbol{x}) = \exp(-\varepsilon(\boldsymbol{x}))/\mathcal{Z}$ with an unknown denominator $\mathcal{Z}$. We make ideal assumptions on the available data $\{\boldsymbol{x}\}$ and energy landscape $\varepsilon(\boldsymbol{x})$; in practice, such ideal assumptions may be approximately fulfilled with *e.g.,* data preprocessing techniques like data augmentation[1].

We define the token index set $\mathbb{L} = \{1, \cdots, L\}$; therefore, $\boldsymbol{x} \equiv \boldsymbol{x}_{\mathbb{L}}$ and $q(\boldsymbol{x})$ is now interpreted as a *joint* PDF. We use $\mathbb{S}, \mathbb{T}$ to denote random subsets of $\mathbb{L}$, where $\mathbb{S} \subset \mathbb{L}$, $\mathbb{T} \subseteq \mathbb{L}$, $\mathbb{T} \neq \emptyset$, and $\mathbb{S} \cap \mathbb{T} = \emptyset$. For simplified notations, we use $q(\boldsymbol{x}_{\mathbb{S}})$ and $q(\boldsymbol{x}_{\mathbb{T}}|\boldsymbol{x}_{\mathbb{S}})$ to denote the $\mathbb{S}$-*marginal* and $\mathbb{T}|\mathbb{S}$-*conditional* PDF of $q(\boldsymbol{x})$, respectively. The PDF of random subsets $(\mathbb{S}, \mathbb{T})$ is denoted as $\rho(\mathbb{S}, \mathbb{T})$.

We use $p_{\boldsymbol{\theta}}(\boldsymbol{x})$ with parameter $\boldsymbol{\theta}$ to denote the *joint* model PDF and assume the existence of a unique[2] global optimum $\boldsymbol{\theta}^*$ such that $p_{\boldsymbol{\theta}^*}(\boldsymbol{x}) = q(\boldsymbol{x})$. Accordingly, the *marginal* $p_{\boldsymbol{\theta}^*}(\boldsymbol{x}_{\mathbb{S}}) = q(\boldsymbol{x}_{\mathbb{S}})$ and the *conditional* $p_{\boldsymbol{\theta}^*}(\boldsymbol{x}_{\mathbb{T}}|\boldsymbol{x}_{\mathbb{S}}) = q(\boldsymbol{x}_{\mathbb{T}}|\boldsymbol{x}_{\mathbb{S}})$ hold true for *any* $(\mathbb{S}, \mathbb{T})$. This also generalizes to *any* transformed $\boldsymbol{y}$-domain with a transformation $\boldsymbol{y} = g(\boldsymbol{x})$; that is, $p_{\boldsymbol{\theta}^*}(\boldsymbol{y}_{\mathbb{T}}|\boldsymbol{y}_{\mathbb{S}}) = q(\boldsymbol{y}_{\mathbb{T}}|\boldsymbol{y}_{\mathbb{S}})$ for *any* $(\mathbb{S}, \mathbb{T})$. Therefore, we say $\boldsymbol{\theta}^*$ indicates the **essence** of $q(\boldsymbol{x})$. The PDF of a random transformation $g(\cdot)$ is denoted as $\tau(g)$.

To present the big cooperative learning in a clarifying and straightforward manner, we additionally assume that one can derive $p_{\boldsymbol{\theta}}(\boldsymbol{y}_{\mathbb{T}}|\boldsymbol{y}_{\mathbb{S}}), \forall (\mathbb{S}, \mathbb{T}, g(\cdot))$ from $p_{\boldsymbol{\theta}}(\boldsymbol{x})$. We will discuss how to address this assumption in practice later.

### 2.2. Conventional Machine Learning Paradigms

Based on the above definitions and notations, we next discuss two popular conventional learning paradigms.

**Maximum Likelihood Learning (MLL)** seeks to maximize the *joint* log-likelihood $\mathbb{E}_{q(\boldsymbol{x})}[\log p_{\boldsymbol{\theta}}(\boldsymbol{x})]$, which is equivalent to minimizing the *joint forward* KL divergence between $q(\boldsymbol{x})$ and $p_{\boldsymbol{\theta}}(\boldsymbol{x})$ (Bishop, 2006; McLachlan & Krishnan, 2007), because

$$\mathbb{E}_{q(\boldsymbol{x})}[-\log p_{\boldsymbol{\theta}}(\boldsymbol{x})] = \mathrm{KL}[q(\boldsymbol{x})||p_{\boldsymbol{\theta}}(\boldsymbol{x})] - C, \quad (1)$$

---

[1]No need for data augmentation if *i.i.d.* data are available.
[2]Equivalent global optima are considered to be the same.

where $C = \mathbb{E}_{q(\boldsymbol{x})}[\log q(\boldsymbol{x})]$ is a constant *w.r.t.* $\boldsymbol{\theta}$. The available information here is data $\{\boldsymbol{x}\}$ from $q(\boldsymbol{x})$; accordingly, Eq. (1) is optimized with Monte Carlo estimation.

Because of the characteristics of forward KL minimization, MLL often suffers from the entrenched challenge associated with strong *mode covering (or zero avoiding)* local optima (Minka et al., 2005); that is, the trained $p_{\boldsymbol{\theta}}(\boldsymbol{x})$ often assigns non-zero densities to where $q(\boldsymbol{x}) \approx 0$, resulting in blurry generated samples (Goodfellow et al., 2014).

**Adversarial Learning** Taking the standard GAN (Goodfellow et al., 2014) as example, adversarial learning parameterizes the model $p_{\boldsymbol{\theta}}(\boldsymbol{x})$ via its generative process $\boldsymbol{x} = G_{\boldsymbol{\theta}}(\boldsymbol{z}), \boldsymbol{z} \sim p(\boldsymbol{z})$, where $G_{\boldsymbol{\theta}}(\cdot)$ is the generator and $p(\boldsymbol{z})$ is an easy-to-sample distribution, and minimizes the *joint* Jensen-Shannon (JS) divergence $\mathrm{JS}[q(\boldsymbol{x})||p_{\boldsymbol{\theta}}(\boldsymbol{x})]$ via

$$\min_{\boldsymbol{\theta}} \max_{\boldsymbol{\beta}} \mathbb{E}_{q(\boldsymbol{x})} \log \sigma[f_{\boldsymbol{\beta}}(\boldsymbol{x})] + \mathbb{E}_{p_{\boldsymbol{\theta}}(\boldsymbol{x})} \log \sigma[-f_{\boldsymbol{\beta}}(\boldsymbol{x})], \quad (2)$$

where $\sigma[\cdot]$ is the sigmoid function. The optimal $f_{\boldsymbol{\beta}^*}(\boldsymbol{x})$ satisfies $f_{\boldsymbol{\beta}^*}(\boldsymbol{x}) = \log q(\boldsymbol{x}) - \log p_{\boldsymbol{\theta}}(\boldsymbol{x})$, which is the negative log density ratio of the reverse KL divergence

$$\mathrm{KL}[p_{\boldsymbol{\theta}}(\boldsymbol{x})||q(\boldsymbol{x})] = \mathbb{E}_{p_{\boldsymbol{\theta}}(\boldsymbol{x})}[\log p_{\boldsymbol{\theta}}(\boldsymbol{x}) - \log q(\boldsymbol{x})]. \quad (3)$$

Since the reverse KL divergence can also be leveraged to form a GAN (Nowozin et al., 2016; Li et al., 2018; Zhao et al., 2020), we use it to represent adversarial learning in this paper, because of its simplicity and natural relationship with the forward-KL-based MLL.

Particularly considering the reverse KL minimization in Eq. (3), the available information could either be $(i)$ data samples $\{\boldsymbol{x}\}$ from $q(\boldsymbol{x})$, where one resorts to Eq. (2) for log density ratio estimation, or $(ii)$ the unnormalized energy landscape $\varepsilon(\boldsymbol{x})$ of $q(\boldsymbol{x})$, which is closely related to sampling Boltzmann distributions (Vargas et al., 2023; Wang et al., 2024b). Different from the forward KL minimization frequently gets stuck in *mode-covering* local optima, the reverse KL minimization suffers from a distinct entrenched challenge related to strong *mode seeking/dropping (or zero forcing)* local optima (Minka et al., 2005; Srivastava et al., 2017); that is, the trained $p_{\boldsymbol{\theta}}(\boldsymbol{x})$ only models limited modes of $q(\boldsymbol{x})$ while ignoring the rest, manifested as mode collapse or insufficient exploration capacity.

Despite their differences, both MLL and adversarial learning share the same commonalities: $(i)$ they both suffer severely from entrenched challenges associated with bad local optima and $(ii)$ they both monotonously exploit the available information in the *joint* space (*e.g.,* all tokens $\{x_l\}_{l=1}^L$ of a sample $\boldsymbol{x} = [x_1, \cdots, x_L]^T$ are always jointly used). Therefore, conventional learning paradigms are mainly about *joint matching* between $q(\boldsymbol{x})$ and $p_{\boldsymbol{\theta}}(\boldsymbol{x})$, which is distinct from cutting-edge foundation models.

## 2.3. Cutting-Edge Foundation Models

Taking shape in the field of natural language processing, foundation models have dramatically changed AI-related research and applications (Devlin et al., 2018; OpenAI, 2022; 2023; Betker et al., 2024; Brooks et al., 2024). Despite their wide range of applications, foundation models share a commonality that the available data information is exploited from diverse perspectives (as detailed below and summarized in Table 1), which is distinctly different from the joint-matching-driven conventional learning paradigms.

**Masked Prediction** Given a universal foundation model $p_{\boldsymbol{\theta}}^{\mathrm{MAE}}(\boldsymbol{x}_{\mathbb{T}}|\boldsymbol{x}_{\mathbb{S}}), \forall(\mathbb{S}, \mathbb{T})$ that satisfies the conditional independence assumption $p_{\boldsymbol{\theta}}^{\mathrm{MAE}}(\boldsymbol{x}_{\mathbb{T}}|\boldsymbol{x}_{\mathbb{S}}) = \prod_{t \in \mathbb{T}} p_{\boldsymbol{\theta}}^{\mathrm{MAE}}(x_t|\boldsymbol{x}_{\mathbb{S}})$, the masked prediction (also termed masked language modeling or masked auto-encoding) seeks to optimize $\boldsymbol{\theta}$ via

$$\max_{\boldsymbol{\theta}} \mathbb{E}_{q(\boldsymbol{x})\rho(\mathbb{S})} \log p_{\boldsymbol{\theta}}^{\mathrm{MAE}}(\boldsymbol{x}_{\mathbb{S}^{\complement}}|\boldsymbol{x}_{\mathbb{S}}), \quad (4)$$

where $\mathbb{S}^{\complement}$ is the complement of $\mathbb{S}$. Often $p_{\boldsymbol{\theta}}^{\mathrm{MAE}}(x_t|\boldsymbol{x}_{\mathbb{S}})$ is modeled as a Categorical PDF[3] for discrete (text) token $x_t \in \mathbb{Z}^{1 \times 1}$ (Devlin et al., 2018) and a Gaussian PDF for continuous (image) token $x_t \in \mathbb{R}^{1 \times D}$ (He et al., 2021).

When a specific $\mathbb{S}$ is of interest, the objective in Eq. (4) equivalently recovers $\mathrm{KL}[q(\boldsymbol{x}_{\mathbb{S}^{\complement}}|\boldsymbol{x}_{\mathbb{S}})||p_{\boldsymbol{\theta}}^{\mathrm{MAE}}(\boldsymbol{x}_{\mathbb{S}^{\complement}}|\boldsymbol{x}_{\mathbb{S}})]$, *i.e.,* the $\mathbb{S}^{\complement}|\mathbb{S}$-*conditional* matching between $q(\boldsymbol{x})$ and $p_{\boldsymbol{\theta}}^{\mathrm{MAE}}(\boldsymbol{x})$. Therefore, the masked prediction exploits the available data information from diverse *conditional* perspectives to form a multi-task training, which averages over all $\mathbb{S}^{\complement}|\mathbb{S}$-conditional matching with weights defined by $\rho(\mathbb{S})$ (see Table 1).

**Next-Token Prediction** Given (often text) data $\{\boldsymbol{x}\}$ from the underlying $q(\boldsymbol{x})$ and a universal auto-regressive foundation model $p_{\boldsymbol{\theta}}^{\mathrm{AR}}(x_t|\boldsymbol{x}_{<t}), \forall t \in \mathbb{L}$, the next-token prediction (also termed auto-regressive/causal language modeling) (Radford et al., 2019) optimizes the parameter $\boldsymbol{\theta}$ with

$$\max_{\boldsymbol{\theta}} \mathbb{E}_{q(\boldsymbol{x})} \frac{1}{L} \sum_{t=1}^L \log p_{\boldsymbol{\theta}}^{\mathrm{AR}}(x_t|\boldsymbol{x}_{<t}), \quad (5)$$

where $\{<t\} \equiv \{1, \cdots, t-1\}$ and thus $\boldsymbol{x}_{<t}$ contains all the tokens prior to the $t$-th token $x_t$.

When a specific $t$ is of interest, the objective in Eq. (5) equivalently recovers $\mathrm{KL}[q(\boldsymbol{x}_{\mathbb{T}}|\boldsymbol{x}_{\mathbb{S}})||p_{\boldsymbol{\theta}}^{\mathrm{AR}}(\boldsymbol{x}_{\mathbb{T}}|\boldsymbol{x}_{\mathbb{S}})]$ with $\mathbb{T} = \{t\}$ and $\mathbb{S} = \{<t\}$. Accordingly, the next-token prediction also exploits the available data information from diverse *conditional* perspectives, albeit from a different set, to form a multi-task training that uniformly averages over all conditional matching associated with next-token prediction.

Although many variants have been proposed to generalize masked prediction and next-token prediction (Yang et al., 2019; Wei et al., 2021; Tian et al., 2024), these two are

---

[3] $-\log p_{\boldsymbol{\theta}}^{\mathrm{MAE}}(x_t|\boldsymbol{x}_{\mathbb{S}})$ recovers the cross-entropy loss.

the most representative. To summarize, existing foundation models make flexible use of the available data information from diverse *conditional* perspectives, distinctly different from conventional learning paradigms that monotonously exploits the available information via *joint matching*. We next generalize on these differences to propose our big cooperative learning that conquers the entrenched local-optima challenges of conventional learning paradigms.

## 3. Big Cooperative Learning

We make ideal assumptions on both the available information and the model capacity (see Section 2.1) such that we can focus on the application-agnostic learning perspective to present our big cooperative learning in a clarifying and straightforward manner. Below, we first reveal that the available information of $q(\boldsymbol{x})$ can be flexibly exploited from diverse perspectives, a portion of which are employed by existing foundation models. We then condense their learning essence and generalize it into the presented big cooperative learning. Finally, tailored simulations with 2-D demonstrable objectives are designed to explicitly justify the principle of big cooperative learning.

### 3.1. Versatile but Underutilized Exploitations of the Available Information

We begin with the most popular situations where the available information of $q(\boldsymbol{x})$ is manifested as *i.i.d.* data samples $\{\boldsymbol{x}\}$. After that, we discuss where the available information is an unnormalized energy landscape $\varepsilon(\boldsymbol{x})$ of $q(\boldsymbol{x})$.

When given a *joint* data sample $\boldsymbol{x} \sim q(\boldsymbol{x})$, one simultaneously receives plenty of versatile data-sampling demonstrations, which include the *joint* $\boldsymbol{x}$ itself, all *marginal* samples $\boldsymbol{x}_{\mathbb{S}}, \forall \mathbb{S}$ (one $\boldsymbol{x}_{\mathbb{S}} \sim q(\boldsymbol{x}_{\mathbb{S}})$ per $\mathbb{S}$), massive *conditional* samples $\boldsymbol{x}_{\mathbb{T}} | \boldsymbol{x}_{\mathbb{S}}, \forall (\mathbb{S}, \mathbb{T})$ (one $\boldsymbol{x}_{\mathbb{T}} \sim q(\boldsymbol{x}_{\mathbb{T}} | \boldsymbol{x}_{\mathbb{S}})$ per $(\mathbb{S}, \mathbb{T})$), and their corresponding counterparts in *any* transformed domain $\boldsymbol{y} = g(\boldsymbol{x}), \forall g(\cdot)$. Note that even an incomplete sample delivers plenty of versatile data-sampling demonstrations.

These readily accessible versatile data-sampling demonstrations, when accumulated across all the available data $\{\boldsymbol{x}\}$ from $q(\boldsymbol{x})$, actually constitute versatile training datasets representing diverse $q(\boldsymbol{y}_{\mathbb{T}} | \boldsymbol{y}_{\mathbb{S}}), \forall (\mathbb{S}, \mathbb{T}, g(\cdot))$, all of which are *different* outward manifestations of the *same unique* essence of $q(\boldsymbol{x})$ (or alternatively $\boldsymbol{\theta}^*$ because $p_{\boldsymbol{\theta}^*}(\boldsymbol{x}) = q(\boldsymbol{x})$). Accordingly, with $D[\cdot||\cdot]$ denoting a specific divergence/distance of PDFs, it's expected that massive objectives $\mathcal{D}[p_{\boldsymbol{\theta}}(\boldsymbol{y}_{\mathbb{T}} | \boldsymbol{y}_{\mathbb{S}})||q(\boldsymbol{y}_{\mathbb{T}} | \boldsymbol{y}_{\mathbb{S}})]$ for various $(\mathbb{S}, \mathbb{T}, g(\cdot))$ shall have *different* local optima but share the *same* global optimum $\boldsymbol{\theta}^*$. Note these versatile datasets of $q(\boldsymbol{y}_{\mathbb{T}} | \boldsymbol{y}_{\mathbb{S}})$, $\forall (\mathbb{S}, \mathbb{T}, g(\cdot))$ need not be explicitly constructed in practice.

Although the available data information can be diversely exploited from massive perspectives, it is likely underexploited

*Table 1.* Representative training objectives of foundation models.

| Model | Objective |
|---|---|
| Masked Prediction BERT (Stickland & Murray, 2019) | $\mathbb{E}_{\rho(\mathbb{S})q(\boldsymbol{x})}\mathrm{KL}[q(\boldsymbol{x}_{\mathbb{S}^{\complement}}|\boldsymbol{x}_{\mathbb{S}})||p_{\boldsymbol{\theta}}^{\mathrm{MAE}}(\boldsymbol{x}_{\mathbb{S}^{\complement}}|\boldsymbol{x}_{\mathbb{S}})]$ 
 $\mathbb{S}$: a random 85% subset of $\mathbb{L}$ |
| Masked Prediction MAE (He et al., 2021) | $\mathbb{E}_{\rho(\mathbb{S})q(\boldsymbol{x})}\mathrm{KL}[q(\boldsymbol{y}_{\mathbb{S}^{\complement}}|\boldsymbol{x}_{\mathbb{S}})||p_{\boldsymbol{\theta}}^{\mathrm{MAE}}(\boldsymbol{y}_{\mathbb{S}^{\complement}}|\boldsymbol{x}_{\mathbb{S}})]$ 
 $\mathbb{S}$: a random 25% subset of $\mathbb{L}$ 
 $\boldsymbol{y}_{\mathbb{S}^{\complement}}$: normalized $\boldsymbol{x}_{\mathbb{S}^{\complement}}$ |
| Masked Prediction MaskFeat (Wei et al., 2021) | $\mathbb{E}_{\rho(\mathbb{S})q(\boldsymbol{x})}\mathrm{KL}[q(\boldsymbol{y}_{\mathbb{S}^{\complement}}|\boldsymbol{x}_{\mathbb{S}})||p_{\boldsymbol{\theta}}^{\mathrm{MAE}}(\boldsymbol{y}_{\mathbb{S}^{\complement}}|\boldsymbol{x}_{\mathbb{S}})]$ 
 $\mathbb{S}$: a random $\approx 60\%$ subset of $\mathbb{L}$ 
 $\boldsymbol{y}_{\mathbb{S}^{\complement}}$: HOG-transformed $\boldsymbol{x}_{\mathbb{S}^{\complement}}$ |
| Next-Token Prediction GPTs (OpenAI, 2022; 2023) | $\mathbb{E}_{\nu(t)q(\boldsymbol{x})}\mathrm{KL}[q(x_t|\boldsymbol{x}_{<t})||p_{\boldsymbol{\theta}}^{\mathrm{AR}}(x_t|\boldsymbol{x}_{<t})]$ 
 $\nu(t) = U[1, L]$ |
| Next-Scale Prediction VAR (Tian et al., 2024) | $\mathbb{E}_{\nu(t)q(\boldsymbol{x})}\mathrm{KL}[q(r_t|\boldsymbol{r}_{<t})||p_{\boldsymbol{\theta}}^{\mathrm{AR}}(r_t|\boldsymbol{r}_{<t})]$ 
 $\boldsymbol{r}$: Multi-scale token maps of $\boldsymbol{x}$ 
 $\nu(t) = U[1, L]$ |
| Permutation Language Modeling (Yang et al., 2019) | $\mathbb{E}_{\chi(\boldsymbol{z})\nu(t)q(\boldsymbol{x})}\mathrm{KL}[q(x_{z_t}|\boldsymbol{x}_{\boldsymbol{z}_{<t}})||p_{\boldsymbol{\theta}}^{\mathrm{AR}^{\dagger}}(x_{z_t}|\boldsymbol{x}_{\boldsymbol{z}_{<t}})]$ 
 $\boldsymbol{z}$: a random permutation |
| Big Cooperative Learning | $\mathbb{E}_{\rho(\mathbb{S}, \mathbb{T})\tau(g)q'(\boldsymbol{y}_{\mathbb{S}})}\mathcal{D}[p_{\boldsymbol{\theta}}(\boldsymbol{y}_{\mathbb{T}}|\boldsymbol{y}_{\mathbb{S}})||q(\boldsymbol{y}_{\mathbb{T}}|\boldsymbol{y}_{\mathbb{S}})]$ 
 $g(\cdot)$: a random transformation 
 $\boldsymbol{y} = g(\boldsymbol{x})$: $g$-transformed $\boldsymbol{x}$ |

in current research. As aforementioned, conventional machine learning paradigms often monotonously exploit it in the *joint* space (see Eqs. (1) and (3)), while most foundation models make use of the available data information from a specific set of *conditional* perspectives (see Eqs. (4) and (5)). Some foundation models have also exploited the available information in domain-knowledge-inspired transformed domains (He et al., 2021; Wei et al., 2021; Tian et al., 2024). For the sake of clarity, we summarize in Table 1 representative training objectives of foundation models and rewrite them in a consistent manner for comparison. It is evident that existing foundation models only exploit the available data information from a limited number of perspectives.

Based on Table 1, we condense the leaning essence of existing foundation models and generalize it into the big cooperative learning presented below, which flexibly contains most foundation-model objectives as special cases and *optionally* exploits the available information from massive perspectives in an exhaustive manner.

Before presenting our big cooperative learning, we first discuss its second main application scenarios, where the available information is manifested as an unnormalized energy landscape $\varepsilon(\boldsymbol{x})$ of $q(\boldsymbol{x})$. Similar to where the available information is data samples $\{\boldsymbol{x}\}$, the energy landscape $\varepsilon(\boldsymbol{x})$ is also underexploited in conventional learning paradigms: the *joint* reverse KL minimization in Eq. (3) monotonously exploits $\varepsilon(\boldsymbol{x})$ in the *joint* space and frequently gets stuck in mode-seeking local optima (Minka et al., 2005; Srivastava et al., 2017). In fact, the unnormalized energy landscape $\varepsilon(\boldsymbol{x})$ can also be exploited from diverse perspec-

tives. For example, when given a *joint* $\varepsilon(\boldsymbol{x})$ such that $q(\boldsymbol{x}) = \exp(-\varepsilon(\boldsymbol{x}))/\mathcal{Z}$, one simultaneously receives all the *conditional* energy landscapes of $q(\boldsymbol{x}_{\mathbb{S}^{\complement}}|\boldsymbol{x}_{\mathbb{S}}), \forall \mathbb{S}$, as well as their corresponding counterparts $q(\boldsymbol{y}_{\mathbb{S}^{\complement}}|\boldsymbol{y}_{\mathbb{S}})$ in *any* monotonically transformed domain with $\boldsymbol{y} = g(\boldsymbol{x})$. For certain applications, the *marginal* energy landscapes of $q(\boldsymbol{y}_{\mathbb{S}}), \forall(\mathbb{S}, g(\cdot))$ may also be accessible. The tailored simulations in Section 3.3 explicitly demonstrate that big cooperative learning diversely exploits the available information of $\varepsilon(\boldsymbol{x})$ to conquer the entrenched mode-seeking local optima challenge.

### 3.2. Big Cooperative Learning with Versatile Exploitations of the Available Information

Noticing the existence of versatile but underutilized exploitations of the available information (either data samples $\{\boldsymbol{x}\}$ or an unnormalized energy landscape $\varepsilon(\boldsymbol{x})$ of $q(\boldsymbol{x})$), we focus on the application-agnostic learning perspective to propose the big cooperative learning, which optionally exploits the available information from massive perspectives in an exhaustive manner and is generally applicable to multiple conventional learning paradigms.

**Definition 3.1** (**Big cooperative learning**). Based on the ideal assumptions of the available information (*i.e.,* data samples $\{\boldsymbol{x}\}$ or an unnormalized energy landscape $\varepsilon(\boldsymbol{x})$ of $q(\boldsymbol{x})$) and the model capacity of $p_{\boldsymbol{\theta}}(\boldsymbol{x})$ in Section 2.1, the **big cooperative learning (abbr. big learning)** trains the model parameter $\boldsymbol{\theta}$ towards the global optimum $\boldsymbol{\theta}^*$ in a massive multi-task cooperative manner, by minimizing

$$\mathbb{E}_{\rho(\mathbb{S},\mathbb{T})\tau(g)q'(\boldsymbol{y}_{\mathbb{S}})}\mathcal{D}[p_{\boldsymbol{\theta}}(\boldsymbol{y}_{\mathbb{T}}|\boldsymbol{y}_{\mathbb{S}})||q(\boldsymbol{y}_{\mathbb{T}}|\boldsymbol{y}_{\mathbb{S}})], \quad (6)$$

where $\rho(\mathbb{S},\mathbb{T}), \tau(g)$ and $q'(\boldsymbol{y}_{\mathbb{S}})$ are user-defined PDFs of random subsets $(\mathbb{S},\mathbb{T})$, a random transformation $\boldsymbol{y} = g(\boldsymbol{x})$, and $\boldsymbol{y}_{\mathbb{S}}$, respectively. $\mathcal{D}[\cdot||\cdot]$ is a divergence/distance metric of PDFs shared across all $(\mathbb{S},\mathbb{T},g(\cdot))$-tasks. It's often convenient to estimate $\mathbb{E}_{\rho(\mathbb{S},\mathbb{T})\tau(g)q'(\boldsymbol{y}_{\mathbb{S}})}[\cdot]$ with one Monte Carlo sample; that is, one task at a time.

*Remark* 3.2 (**Task diversity**). The task diversity is defined by $\rho(\mathbb{S},\mathbb{T}), \tau(g), q'(\boldsymbol{y}_{\mathbb{S}})$, and $\mathcal{D}[\cdot||\cdot]$. Since both $\mathbb{S} = \emptyset$ and $\mathbb{T} = \mathbb{L}$ are possible, Eq. (6) could exhaustively cover all *joint*, *marginal*, and *conditional* matching tasks across many $g(\cdot)$-transformed domains. Note certain tasks also enable exploiting incomplete data (or energy landscapes). One metric $\mathcal{D}[\cdot||\cdot]$ is shared across all $(\mathbb{S},\mathbb{T},g(\cdot))$-tasks, because (*i*) metrics can conflict (Minka et al., 2005; Zhao et al., 2020) and (*ii*) $\mathcal{D}[\cdot||\cdot]$ is application dependent and often determined by how the available information is manifested. For example, it's often convenient to set $\mathcal{D}[\cdot||\cdot]$ as the forward or reverse KL divergence if the available information is manifested as data samples or an energy landscape, respectively.

*Remark* 3.3 (**Task cooperation**). As discussed previously, all tasks $\mathcal{D}[p_{\boldsymbol{\theta}}(\boldsymbol{y}_{\mathbb{T}}|\boldsymbol{y}_{\mathbb{S}})||q(\boldsymbol{y}_{\mathbb{T}}|\boldsymbol{y}_{\mathbb{S}})]$ for various $(\mathbb{S},\mathbb{T},g(\cdot))$ have *different* local optima but share the *same* global optimum $\boldsymbol{\theta}^*$. Therefore, if one gets stuck in a local optimum

$\bar{\boldsymbol{\theta}}_A$ of Task A, doing another Task B (of which $\bar{\boldsymbol{\theta}}_A$ is not a local optimum) would help Task A jump out of the local optimum, demonstrating cooperation. Accordingly, the big cooperative learning randomly switching among its massive tasks (via one-sample-based Monte Carlo estimation of Eq. (6)) to leverage cooperation among tasks to conquer their *inconsistent* local optima, while concurrently encouraging exploring the *consistent* global optimum. Note that no task competition/conflict is expected at the global optimum.

Considering practical applications, it's essential that the tasks of big cooperative learning are diverse enough to contain a "Task B" for that "Task A", which guarantees a probability in conquering the associated local optimum. Note that, even if the task diversity is not sufficient, big cooperative learning is also expected to find an improved local optimum, which is a "global optimum" *w.r.t.* the employed task scope.

*Remark* 3.4 (**Modeling of $p_{\boldsymbol{\theta}}(\cdot)$**). To focus on investigating the learning of $p_{\boldsymbol{\theta}}(\cdot)$, we have made ideal assumptions on the orthogonal dimension of its modeling in Section 2.1, *i.e.,* one can access analytic $p_{\boldsymbol{\theta}}(\boldsymbol{y}_{\mathbb{T}}|\boldsymbol{y}_{\mathbb{S}}), \forall(\mathbb{S},\mathbb{T},g(\cdot))$. In this paper, we leverage GMMs, a universal approximator of densities (Lindsay, 1995; Peel & MacLahlan, 2000; Goodfellow et al., 2016), and random *orthogonal* transformations $\boldsymbol{y} = g(\boldsymbol{x})$ to fulfill those assumptions while keeping representativeness. Considering applying big cooperative learning to where DNNs are of interest, we reveal that one can follow existing foundation models to leverage a universal DNN-based model to *simultaneously approximate* $p_{\boldsymbol{\theta}}(\boldsymbol{y}_{\mathbb{T}}|\boldsymbol{y}_{\mathbb{S}})$s for all $(\mathbb{S},\mathbb{T},g(\cdot))$s of interest. Note that DNN is also a universal approximator of PDFs (Lu & Lu, 2020). However, how to ensure the interrelationship (*i.e.,* Bayes' rule) among different $p_{\boldsymbol{\theta}}(\boldsymbol{y}_{\mathbb{T}}|\boldsymbol{y}_{\mathbb{S}})$s is worth future research.

*Remark* 3.5 (**Multi-modal generalization**). By interpreting paired multi-modal data $(\boldsymbol{a}, \boldsymbol{b}, \boldsymbol{c})$ as a *joint* sample $\boldsymbol{x} = [\boldsymbol{a}, \boldsymbol{b}, \boldsymbol{c}]$, the big cooperative learning in Definition 3.1 can be leveraged to handle multi-modal applications.

### 3.3. Tailored 2-Dimensional Simulations to Explicitly Demonstrate the Big-Learning Principle

To explicitly verify the big cooperative learning, one would expect a situation where (*i*) both the available information and the model capacity satisfy the ideal assumptions in Section 2.1 so as to focus on the learning for investigation, (*ii*) both the local and global optima are readily interpretable and they can be flexibly controlled, (*iii*) conventional learning paradigms suffer from entrenched local-optima challenges, and (*iv*) each task objective $\mathcal{D}[p_{\boldsymbol{\theta}}(\boldsymbol{y}_{\mathbb{T}}|\boldsymbol{y}_{\mathbb{S}})||q(\boldsymbol{y}_{\mathbb{T}}|\boldsymbol{y}_{\mathbb{S}})]$ of Eq. (6) is a 2-dimensional (2-D) demonstrable function such that one need not consider the influence of optimization.

With that in mind, we bypass black-box DNNs and leverage GMMs to design tailored simulations that satisfy all the aforementioned conditions. Specifically, we set $\mathcal{D}[\cdot||\cdot]$ as the

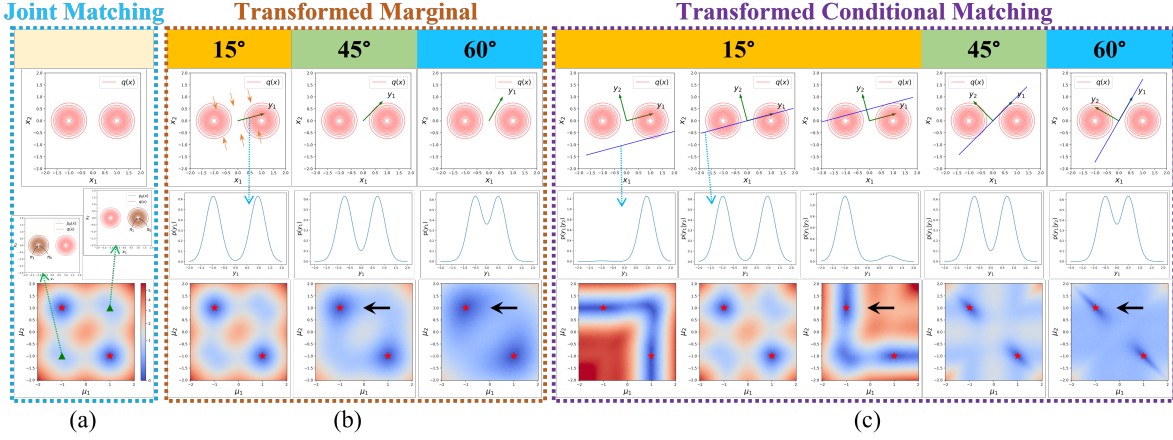

Figure 1. Explicit demonstration of the principle of big cooperative learning with tailored 2-D simulations. The first row generally indicates the experimental setup, such as $q(\boldsymbol{x})$ or the transformed space. The second row shows the local optima or the exploited $q(\cdot)$-information, *i.e.,* the energy landscape of $q(y_1)/q(y_1|y_2)$. The third row demonstrates the 2-D objective surfaces.

*reverse* KL divergence[4]. The available information is then set as an unnormalized energy landscape $\varepsilon(\boldsymbol{x})$ of the underlying $q(\boldsymbol{x}) = \sum_{i=1}^{2} \frac{1}{2}\mathcal{N}(\boldsymbol{x}|\boldsymbol{\mu}_i^*, \sigma^2\mathbf{I})$ where $\boldsymbol{\mu}_1^* = [-1, 0]^T$, $\boldsymbol{\mu}_2^* = [1, 0]^T$, and $\sigma^2$ is a hyperparameter; see Fig. 1a. To enable 2-D demonstrable objectives, we employ a tailored modeling[5] of $p_{\boldsymbol{\theta}}(\boldsymbol{x}) = \sum_{i=1}^{2} \frac{1}{2}\mathcal{N}(\boldsymbol{x}|\boldsymbol{\mu}_i, \sigma^2\mathbf{I})$, where $\boldsymbol{\mu}_1 = [\mu_1, 0]^T$, $\boldsymbol{\mu}_2 = [\mu_2, 0]^T$, and $\boldsymbol{\theta} = [\mu_1, \mu_2]^T$ is the 2-D parameter. Accordingly, each task objective $\mathrm{KL}[p_{\boldsymbol{\theta}}(\cdot)||q(\cdot)]$ is a 2-D demonstrable function of $\boldsymbol{\theta}$.

The objective of the conventional *joint* matching, *i.e.,* $\mathrm{KL}[p_{\boldsymbol{\theta}}(\boldsymbol{x})||q(\boldsymbol{x})]$, is explicitly demonstrated in Fig. 1a, where it's apparent that two strong *mode-seeking* local optima emerge (marked as green triangles).

Regarding the big cooperative learning, we employ random rotational transformations $\boldsymbol{y} = g(\boldsymbol{x}) = \mathbf{A}\boldsymbol{x}$ here for simplicity, where $\mathbf{A}$ is a random rotation matrix, and explicitly demonstrate in Figs. 1b and 1c its sample task objectives of (*i*) transformed *marginal* matching $\mathrm{KL}[p_{\boldsymbol{\theta}}(y_1)||q(y_1)]$ with $\mathbb{T} = \{1\}$, $\mathbb{S} = \emptyset$, and $\mathbf{A}$ denoting $15°$, $45°$, and $60°$ rotations, respectively, and (*ii*) transformed *conditional* matching $\mathrm{KL}[p_{\boldsymbol{\theta}}(y_1|y_2)||q(y_1|y_2)]$ with $\mathbb{T} = \{1\}$, $\mathbb{S} = \{2\}$, and the same set of rotations. It's evident that different tasks of big cooperative learning have *different* local optima but share the *same* global optimum (marked as red stars). More importantly, for a specific Task A of the joint matching, its strong local optimum (*e.g.,* the north-east green triangle) can be readily conquered by many potential Task Bs, as indicated by the black arrows, demonstrating the potential of cooperation among tasks. We further investigate the relationships between different exploitations of the available infor-

mation and the local-optima patterns of the correspondingly objective. By parallel comparing the second and third rows of Fig. 1b, it is apparent that as the intersection of the two modes increases, the local optima of transformed marginal matching gradually vanish, albeit at the cost of a decreased sharpness around the global optimum; similar phenomena are observed in Fig. 1c, which further reveal the potential of cooperation among tasks. Therefore, by randomly switching among these tasks, big cooperative learning is expected to form cooperation among them to destabilize and conquer their *inconsistent* local optima, while concurrently encourage exploring the *same* global optimum.

The above tailored 2-D simulations explicitly justify the principle of big cooperative learning within the RKL territory in a lightweight manner. In the experiments, we will leverage more challenging simulations on both forward and reverse KL minimizations to demonstrate the effectiveness and, more importantly, the emerging power of exploration of our big cooperative learning.

## 4. Related Work

**Multi-Task Learning** trains a model from multiple related tasks simultaneously (Caruana, 1997; Ruder, 2017; Zhang & Yang, 2021; Chen et al., 2024a) for improved performance, generalization, robustness to data sparsity, *etc.* In a broad sense, our big cooperative learning falls into the category of multi-task learning.

However, classical multi-task learning concentrates on "external" knowledge transfer among *several* related but potentially distinct tasks, which *e.g.,* possess different supervision, modalities, and/or even goals (Nishino et al., 2019; Zhang & Yang, 2021; Hu et al., 2024; Zhang et al., 2024; Chen et al., 2024a; Xu et al., 2023). These tasks are often heuristically

---

[4]One needs $\geq 3$ GMM components to simulate the local optima of the *forward* KL divergence (Jin et al., 2016; Chen et al., 2024b).

[5]For a 2-component GMM, the dimensionality of $\boldsymbol{\theta}$ is in general 11, where the objective is not easy to demonstrate.

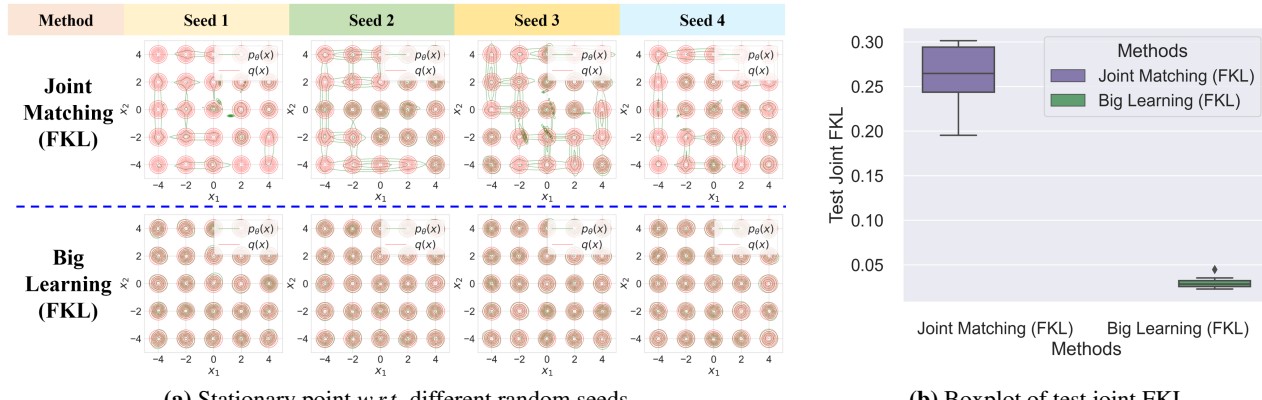

(a) Stationary point *w.r.t.* different random seeds

(b) Boxplot of test joint FKL

*Figure 2.* Big cooperative learning conquers the mode-covering local optima challenge of the conventional FKL-based joint matching.

assembled without solid theoretical support; accordingly, task competition/conflict (Ruder, 2017) frequently emerges, *i.e.,* different tasks compete for the model's capacity, resulting in poor performance on certain tasks.

By comparison, our big cooperative learning, being markedly different, focuses on "internal" information exploitations through *massive cooperative* tasks, which are designed by diversely exploiting the available information of $q(\boldsymbol{x})$ from versatile perspectives. Accordingly, the *massive* tasks of big cooperative learning share the same global optimum, where no task competition/conflict is expected.

**Diverse Information Exploitation** The idea of diversely exploiting the available data information underlies a lot of AI-related research, such as the foundation models revealed in Section 3.1, neural processes (Garnelo et al., 2018a;b; Kim et al., 2019; Nguyen & Grover, 2022; Shih et al., 2022; Maraval et al., 2024), and other related research (Bao et al., 2023; Cong & Li, 2024). Often relatively limited use is made of the available data information (*e.g.,* via diverse conditional matching in the original domain) to deliver a specific output (such as pretraining or diverse conditional data sampling capabilities).

In contrast, we reveal and unify the principle of diverse information exploitation from a general application-agnostic learning perspective to present the big cooperative learning concept, which is generally applicable to situations where the available information is manifested as data samples or unnormalized energy landscapes. More importantly, we leverage tailored simulations to explicitly and empirically prove that big cooperative learning delivers a more general output of conquering local optima and encouraging the exploration of the global optimum.

## 5. Experiments

To focus on the application-agnostic learning perspective to verify the big cooperative learning in a clarifying and

straightforward manner, we leverage flexible GMMs to design challenging simulations, where the ideal assumptions in Section 2.1 are satisfied. Since a GMM is a universal approximator of densities, the GMM-based simulation setup is not deemed particularly restrictive.

Specifically, we set the underlying $q(\boldsymbol{x})$ as a GMM with $K$ components and employ a perfect modeling of $p_{\boldsymbol{\theta}}(\boldsymbol{x})$, *i.e.,*

$$q(\boldsymbol{x}) = p_{\boldsymbol{\theta}^*}(\boldsymbol{x}) = \sum\nolimits_{k=1}^{K} \pi_k^* \mathcal{N}(\boldsymbol{x}|\boldsymbol{\mu}_k^*, \boldsymbol{\Sigma}_k^*)$$
$$p_{\boldsymbol{\theta}}(\boldsymbol{x}) = \sum\nolimits_{k=1}^{K} \pi_k \mathcal{N}(\boldsymbol{x}|\boldsymbol{\mu}_k, \boldsymbol{\Sigma}_k),$$
(7)

where $K = 25$, $\pi_k^* = 1/K$, $\boldsymbol{\mu}_k^*$s are placed on a grid (see Fig. 2a), and $\boldsymbol{\Sigma}_k^* = \sigma^2 \mathbf{I}$ with hyperparameter $\sigma^2$. Other details are given in Appendices B and C.

To demonstrate the general effectiveness of the big cooperative learning in addressing entrenched local-optima challenges across multiple conventional learning paradigms, we design GMM-based simulations on both mode-covering forward KL (FKL) minimization (associated with where the available information is data samples $\{\boldsymbol{x}\}$ from $q(\boldsymbol{x})$) and mode-seeking reverse KL (RKL) minimization (related to where the available information is an unnormalized energy landscape $\varepsilon(\boldsymbol{x})$ of $q(\boldsymbol{x})$). To our knowledge, the big cooperative learning is the first research that *simultaneously* conquers the local optima of both FKL and RKL minimizations in an elegant way (*i.e.,* only diverse exploitations of the available information are additionally employed).

### 5.1. Big Cooperative Learning Conquers Local Optima of Forward KL Minimization

We first demonstrate the effectiveness of big cooperative learning on the *mode covering* FKL minimization. We make empirical comparisons between $(i)$ *joint matching* with the joint FKL objective $\mathrm{KL}[q(\boldsymbol{x})\|p_{\boldsymbol{\theta}}(\boldsymbol{x})]$, which is equivalent to the conventional maximum likelihood learning, and $(ii)$ the big cooperative learning with FKL-based objective

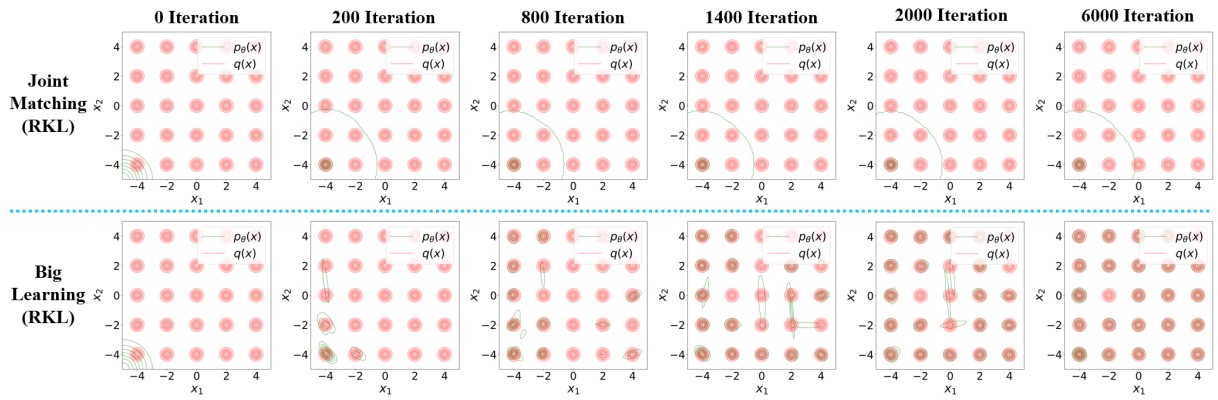

*Figure 3.* Big cooperative learning endows the mode-seeking RKL-based joint matching with power of exploration.

$\mathbb{E}_{\rho(\mathbb{S},\mathbb{T})\tau(g)q'(\boldsymbol{y}_{\mathbb{S}})}\text{KL}[q(\boldsymbol{y}_{\mathbb{T}}|\boldsymbol{y}_{\mathbb{S}})||p_{\boldsymbol{\theta}}(\boldsymbol{y}_{\mathbb{T}}|\boldsymbol{y}_{\mathbb{S}})]$, where $\rho(\mathbb{S},\mathbb{T})$, $\tau(g)$, and $q'(\boldsymbol{y}_{\mathbb{S}})$ are specified to include the *joint*, all *marginal*, and random orthogonally *transformed marginal* matching tasks. During learning, only the available information (*i.e., i.i.d.* data samples $\{\boldsymbol{x}\}$) of $q(\boldsymbol{x})$ is used for both methods. Note the major difference of big cooperative learning is its diverse exploitations of the same data $\{\boldsymbol{x}\}$.

Experimental results are summarized in Fig. 2, where it's expected that the conventional *joint matching* suffers severely from mode-covering local optima that demonstrate both "one-fits-many" and "many-fit-one" patterns (Chen et al., 2024b). By comparison, the FKL-based big cooperative learning stably delivers the global optimum in this simulation, despite it utilizes the *same* available data information (but with additional diverse exploitations). Therefore, by considering that both methods employ the same mode-covering FKL divergence, it's evident that it is the versatile exploitations of the available information that conquer the entrenched local-optima challenge of the conventional FKL-based joint matching, justifying the effectiveness of big cooperative learning.

### 5.2. Big Learning Endows Reverse KL Minimization With Emerging Power of Exploration

We next leverage GMM-based white-box simulations to reveal that the big cooperative learning can also address the entrenched challenges of the *mode seeking* RKL minimization, with a highlight on its emerging power of exploration.

We make parallel comparisons between the training processes of $(i)$ the conventional learning paradigm of the RKL-based *joint matching*, whose objective is $\text{KL}[p_{\boldsymbol{\theta}}(\boldsymbol{x})||q(\boldsymbol{x})]$, and $(ii)$ the big cooperative learning with objective $\mathbb{E}_{\rho(\mathbb{S},\mathbb{T})\tau(g)q'(\boldsymbol{y}_{\mathbb{S}})}\text{KL}[p_{\boldsymbol{\theta}}(\boldsymbol{y}_{\mathbb{T}}|\boldsymbol{y}_{\mathbb{S}})||q(\boldsymbol{y}_{\mathbb{T}}|\boldsymbol{y}_{\mathbb{S}})]$, where $\rho(\mathbb{S},\mathbb{T})$, $\tau(g)$, and $q'(\boldsymbol{y}_{\mathbb{S}})$ are specified to include the *joint*, *marginal*, *conditional*, and random orthogonally *transformed marginal and conditional* matching tasks. During learning, only the

available information (*i.e.,* the unnormalized energy landscape $\varepsilon(\boldsymbol{x})$) of $q(\boldsymbol{x})$ is used. We deliberately initialize $\{\boldsymbol{\mu}_i\}$s with $\mathcal{N}(-5, 0.01)$ to encourage mode collapse for strengthened challenge (see Fig. 3). Similarly, the big cooperative learning mainly differs in its diverse exploitations of the same available information of $\varepsilon(\boldsymbol{x})$.

Fig. 3 explicitly demonstrates the training processes of both methods. It's evident that the conventional RKL-based *joint matching* suffers severely from mode collapse, showing feeble exploration as expected. By contrast, the RKL-based big cooperative learning manages to deliver a *surprising power of exploration*, even though it uses the same available information and all its tasks are based on the mode-seeking RKL. Therefore, it has to be the versatile exploitations of the available information that conquer the entrenched mode-seeking local-optima challenge of the conventional RKL-based joint matching, justifying the big cooperative learning from another important perspective.

## 6. Concluding Remarks

By summarizing and generalizing the learning of foundation models, we present the general learning concept of big cooperative learning, which diversely exploits the available information in a massive multi-task cooperative manner to address the entrenched local-optima challenges of conventional machine learning paradigms. Tailored GMM-based simulations are carried out to explicitly demonstrate its general effectiveness in simultaneously conquering the local optima of both FKL and RKL minimizations that represent maximum likelihood and adversarial learning, respectively.

Although big cooperative learning is inspired by foundation models, it hasn't provided a solid positive feedback for their improvement. We leave this for future research. Another valuable future research may concern about generalizing big cooperative learning with an optimized task sequence for improved learning efficiency and exploration power.

## Impact Statement

This paper presents work whose goal is to advance the field of Machine Learning. There are many potential societal consequences of our work, none which we feel must be specifically highlighted here.

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

# A. Details and Additional Results of Tailored 2-D Simulations

Given GMM-based $q(\boldsymbol{x})$ and $p_{\boldsymbol{\theta}}(\boldsymbol{x})$ with 2-D parameter $\boldsymbol{\theta} = [\mu_1, \mu_2]^T$, *i.e.,*

$$q(\boldsymbol{x}) = p_{\boldsymbol{\theta}^*}(\boldsymbol{x}) = \sum_{i=1}^{2} \frac{1}{2} \mathcal{N}(\boldsymbol{x}|\boldsymbol{\mu}_i^*, \sigma^2 \mathbf{I})$$

$$p_{\boldsymbol{\theta}}(\boldsymbol{x}) = \sum_{i=1}^{2} \frac{1}{2} \mathcal{N}(\boldsymbol{x}|\boldsymbol{\mu}_i, \sigma^2 \mathbf{I}), \tag{8}$$

and by specifying $\mathcal{D}[\cdot||\cdot]$ as the *reverse* KL divergence, we approximately calculate the 2-D objective of the conventional *joint* matching, *i.e.,* $\mathrm{KL}[p_{\boldsymbol{\theta}}(\boldsymbol{x})||q(\boldsymbol{x})]$, with Monte Carlo estimation using 2000 samples from $p_{\boldsymbol{\theta}}(\boldsymbol{x})$. Accordingly, the *joint* $\mathrm{KL}[p_{\boldsymbol{\theta}}(\boldsymbol{x})||q(\boldsymbol{x})]$ is a 2-D function that can be explicitly demonstrated.

Similarly, $\mathrm{KL}[p_{\boldsymbol{\theta}}(x_l)||q(x_l)], l \in 1, 2$ for *marginal* matching, $\mathrm{KL}[p_{\boldsymbol{\theta}}(x_l|x_m)||q(x_l|x_m)], l \neq m$ for *conditional* matching, $\mathrm{KL}[p_{\boldsymbol{\theta}}(y_l)||q(y_l)], l \in 1, 2$ for *transformed marginal* matching, $\mathrm{KL}[p_{\boldsymbol{\theta}}(y_l|y_m)||q(y_l|y_m)], l \neq m$ for *transformed conditional* matching, can also be approximately calculated and demonstrated as a 2-D image.

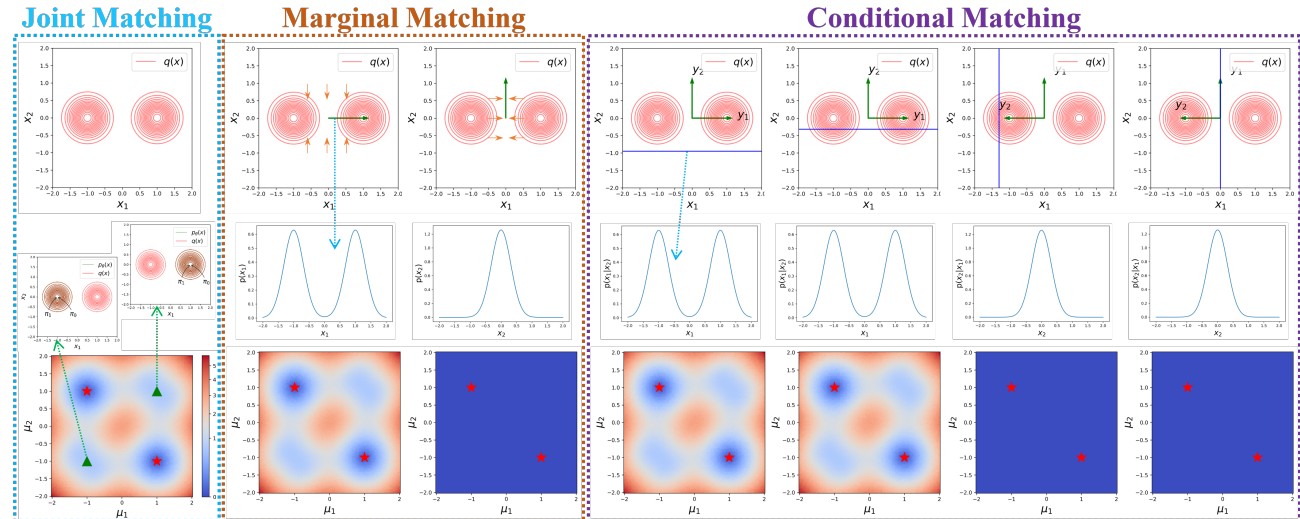

*Figure 4.* Demonstration of the *joint* matching, *marginal* matching, and *conditional* matching in the original domain. The two global optima are marked with red stars. $\sigma^2 = 0.1$.

Fig. 4 shows the *joint* matching, *marginal* matching, and *conditional* matching in the original domain. It's evident that, similar to Fig. 1 of the main manuscript, cooperation does not emerge between *any* two tasks; but, at least, tasks are harmless to each other because they all share the same global optimum. This implies:

- **In situations with independent features/tokens, no cooperation would arise for naive joint, marginal, and conditional matching. Accordingly, existing foundation models likely fail in such situations.**

- **The task scope of big cooperative learning, *i.e.,* the versatility in exploiting the available information, is essential.**

During our investigations in the tailored 2-D simulations, we discover several **interesting side-products** (as summarized below) that potentially benefit implementations of foundation models.

- Big cooperative learning constructed with diverse transformed *joint and marginal* matching may favor a bi-level optimization (*i.e.,* training multiple steps in one matching before moving on to the next), because, as shown in Fig. 5b, direct averaging over diverse marginal matching may not address local optima sufficiently but training multiple steps in one matching (like the 70° plot in Fig. 5a) would conquer local optima.

- Big cooperative learning constructed with diverse transformed *conditional* matching need no bi-level optimization, because direct averaging over diverse transformed conditional matching may already conquer local optima, as shown in Fig. 6. This is akin to what's done in existing foundation models.

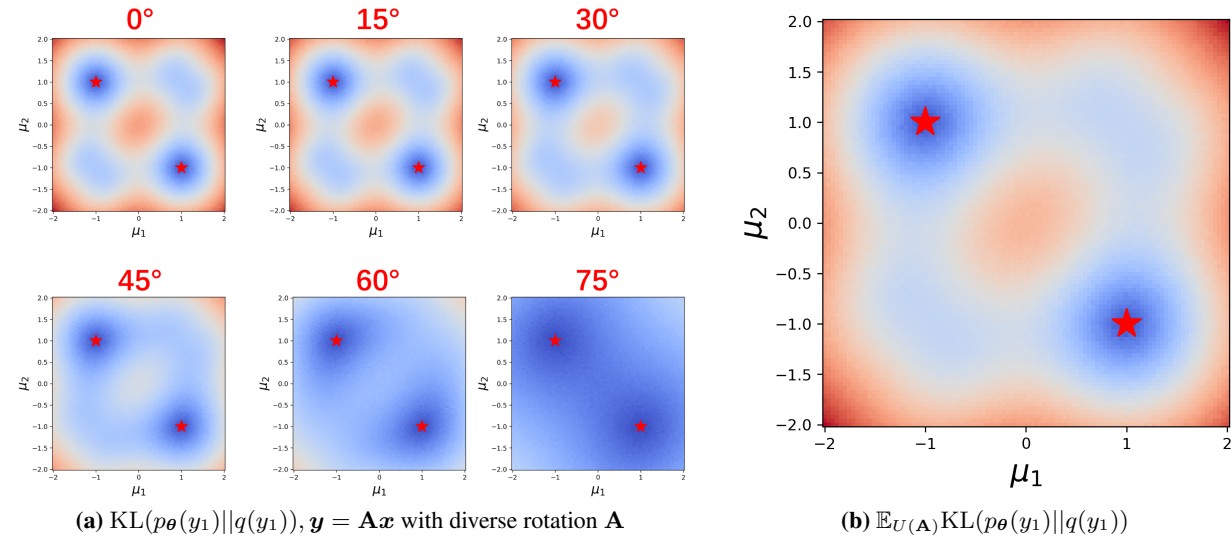

**(a)** $\mathrm{KL}(p_{\boldsymbol{\theta}}(y_1)||q(y_1))$, $\boldsymbol{y} = \mathbf{A}\boldsymbol{x}$ with diverse rotation $\mathbf{A}$  **(b)** $\mathbb{E}_{U(\mathbf{A})}\mathrm{KL}(p_{\boldsymbol{\theta}}(y_1)||q(y_1))$

*Figure 5.* Demonstration of the preference of a bi-level optimization when using transformed joint and marginal matching. (a) Transformed marginal matching has an magnitude correlated with the significance of local optima. (b) Optimization with an uniformly sampled marginal matching may not sufficiently address local optima, where a bi-level optimization would be beneficial. See the supplementary `Figure_Tailored_Simulation_video_margin.gif`.

- Multi-scale noising (when applicable) serves as powerful transformations for big cooperative learning, as illustrated in Fig. 7; this is akin to diffusion models (Ho et al., 2020; Song et al., 2020). It's worth noting that the significance of local optima increases with the decreasing of $\sigma$.

## B. Details on the 25-GMM Simulations on Forward KL Minimization

We use *i.i.d.* samples $\{\boldsymbol{x}\}$ from $q(\boldsymbol{x})$, where the hyperparameter $\sigma^2$ is set to $0.1$, and employ a model with a perfectly matched model capacity, *i.e.,*

$$p_{\boldsymbol{\theta}}(\boldsymbol{x}) = \sum\nolimits_{i=1}^{25} \pi_i \mathcal{N}(\boldsymbol{x}|\boldsymbol{\mu}_i, \boldsymbol{\Sigma}_i), \tag{9}$$

where $\boldsymbol{\theta} = \{\pi_i, \boldsymbol{\mu}_i, \boldsymbol{\Sigma}_i\}_{i=1}^{25}$. $\{\boldsymbol{\mu}_i\}$s are randomly initialized with $\mathcal{N}(0,1)$ for strengthened challenge. We employ the *joint* matching, all *marginal*, and random *transformed marginal* matching to constitute our big cooperative learning. We resort to the Expectation Maximization (EM) for the optimization of each forward KL (FKL) matching, because all the employed FKL matching has analytic EM updates and empirically EM updates are efficient. The random transformation $\boldsymbol{y} = g(\boldsymbol{x})$ is specified as a random orthogonal transformation, *i.e.,* $\boldsymbol{y} = \mathbf{A}\boldsymbol{x}$ with $\mathbf{A}$ generated with `scipy.stats.ortho_group.rvs`.

Following the interesting side-products revealed above, we employ a bi-level optimization, *i.e.,* $5$ training steps are performed in one matching before moving on to the next. Fig. 2 of the main manuscript shows that big cooperative learning exploits the available data information from diverse perspectives to conquer the mode-covering local-optima dilemma of the conventional joint FKL minimization.

## C. Details on the 25-GMM Simulations on Reverse KL Minimization

We resort to Stochastic Gradient Descent (SGD) for the optimization of the *mode seeking* reverse KL (RKL) minimization. We set the hyperparameter $\sigma^2$ of $q(\boldsymbol{x})$ to $0.05$. For simplicity, we parameterize $p_{\boldsymbol{\theta}}(\boldsymbol{x})$ as

$$p_{\boldsymbol{\theta}}(\boldsymbol{x}) = \sum\nolimits_{i=1}^{25} \frac{1}{25} \mathcal{N}(\boldsymbol{x}|\boldsymbol{\mu}_i, \mathbf{L}_i \mathbf{L}_i^T), \tag{10}$$

where $\boldsymbol{\theta} = \{\boldsymbol{\mu}_i, \mathbf{L}_i\}_{i=1}^{25}$ and $\mathbf{L}_i$ is a lower-triangular matrix. $\{\boldsymbol{\mu}_i\}$s are randomly initialized with $\mathcal{N}(-5, 0.01)$ such that all 25 components are initialized to the lower left corner of $q(\boldsymbol{x})$ (see Fig. 3 of the main manuscript); accordingly, the conventional RKL minimization likely suffers from severe mode collapse/seeking (*i.e.,* all 25 components model the one

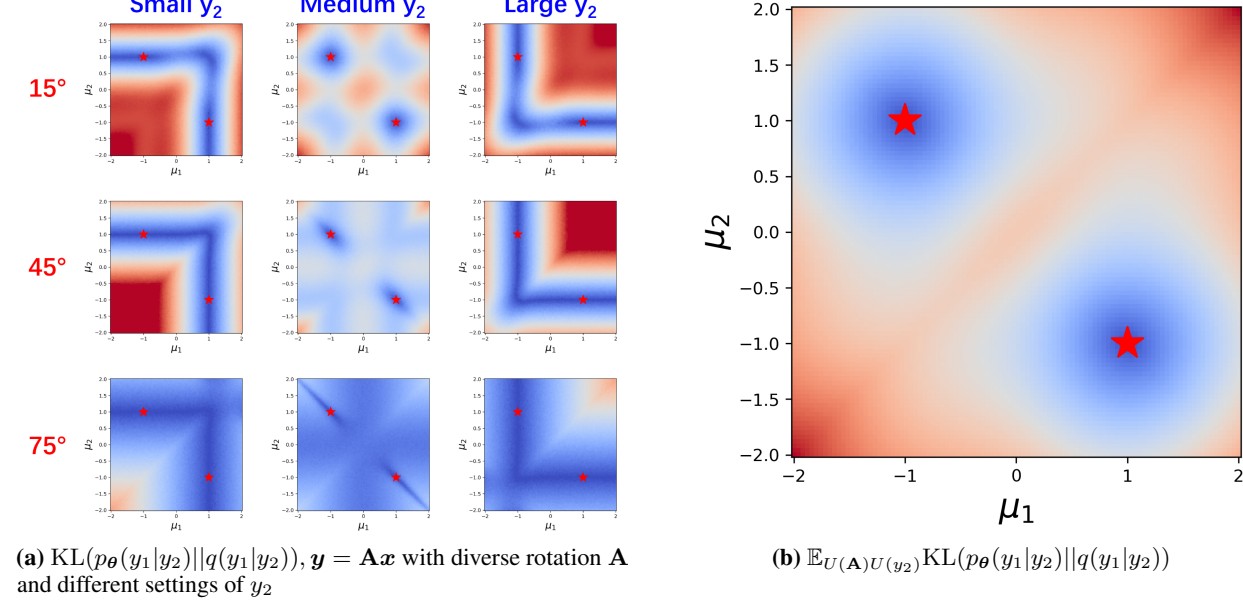

**(a)** $\mathrm{KL}(p_{\boldsymbol{\theta}}(y_1|y_2)||q(y_1|y_2))$, $\boldsymbol{y} = \mathbf{A}\boldsymbol{x}$ with diverse rotation $\mathbf{A}$ and different settings of $y_2$

**(b)** $\mathbb{E}_{U(\mathbf{A})U(y_2)}\mathrm{KL}(p_{\boldsymbol{\theta}}(y_1|y_2)||q(y_1|y_2))$

*Figure 6.* Demonstration of the principle of diverse conditional matching. (a) Transformed conditional matching has a loss surface diversely changing with different rotations and conditions. (b) Optimization with an uniformly sampled conditional matching may already conquer local optima, delivering an appealing averaged loss surface. See the supplementary `Figure_Tailored_Simulation_video_condition.gif`.

lower left component of $q(\boldsymbol{x})$; see Fig. 3 of the main manuscript). Such a challenging initialization highlights the remarkable power of exploration of the presented big cooperative learning.

Our big cooperative learning is specified to include the *joint*, all *marginal*, diverse *conditional*, and random orthogonally *transformed marginal and conditional* matching tasks. For SGD optimization, we use a learning rate of $0.1$, a mini-batch of $100$ (*i.e.,* $100$ samples from $p_{\boldsymbol{\theta}}(\boldsymbol{x})$ are used to calculate a stochastic gradient) Fig. 3 of the main manuscript proves that big cooperative learning delivers remarkable power of exploration that conquers the mode-seeking (or mode-collapse) local-optima dilemma of the conventional joint RKL minimization.

*Figure 7.* Demonstration of the power of multi-scale noising. Multi-scale noising (when applicable) serves as a new dimension for transformations applicable in big cooperative learning. Note the local optima gradually vanish with the increasing noise variance $\sigma^2$, but the local surfaces surrounding the global optimum are also gradually flattened. It's expected that different characteristics among multi-scale noising would deliver cooperation.

