# OpenReview forum: "Big Cooperative Learning to Conquer Local Optima"
_ICML.cc/2025/Conference — Submitted to ICML 2025_

### Official Review · Reviewer_4KZD · 2025-03-10

**Overall Recommendation:** 4

**Summary:**

This paper introduces Big Cooperative Learning (BCL), a strategy to circumvent local optima by exploiting multiple “views” of the same data distribution. Instead of using one global objective, BCL sets up many subtasks (e.g., marginal or conditional matching, or transformations of the features). All tasks share the same global optimum yet have distinct local minima. By randomly switching among tasks, the algorithm “destabilizes” any single task’s local optimum and converges on the global solution that satisfies all tasks simultaneously. Experiments with Gaussian mixtures show that BCL’s multi-task scheme outperforms conventional single-objective methods in avoiding mode collapse or mode-covering issues, illustrating its potential to tackle entrenched local-optima problems in both forward KL (maximum-likelihood) and reverse KL (adversarial) learning settings.

**Claims And Evidence:**

Broadly, the core theoretical claim—that multiple subtasks with a shared global optimum can help escape local minima—is backed by two-dimensional and higher-dimensional Gaussian mixture experiments. These experiments illustrate how BCL consistently converges on the true parameters in scenarios where single-objective methods often fail. In that sense, the authors do provide clear and convincing evidence for the mechanism at work within those controlled mixture-model settings.

However, generalization to large-scale neural networks or “foundation model” training remains less explored. The paper does connect BCL’s multi-task idea to how modern foundation models use diverse objectives (e.g., masked or conditional predictions), but it stops short of offering the same level of direct empirical evidence in real neural architectures. Further, theoretical evidence proving that BCL always overcomes local optima under broader conditions is lacking, so while the main demonstration (that BCL escapes local minima in mixture models) is well supported, the broader implication—that it universally conquers local optima—would still need more thorough theoretical and large-scale empirical validation.

**Essential References Not Discussed:**

NA

**Experimental Designs Or Analyses:**

Strengths:

Clear Problem & Setup: Effectively defines the local optima problem and motivates their solution.

Controlled GMM Simulations: GMMs provide a valid, well-understood environment to isolate the method's effects.

Targeted Demonstrations: Simulations effectively show the method's ability to handle local optima (forward KL) and mode collapse (reverse KL).

Comprehensive Task Design: Explores joint, marginal, conditional, and transformed matching, supporting conclusions about task diversity.

Concerns:

Idealized Assumptions: Strong assumptions about data/model capacity limit real-world applicability.

Limited Model Complexity: GMMs are simpler than DNNs; findings need further validation with DNNs.

Lack of Statistical Rigor: Could benefit from statistical significance tests.

**Methods And Evaluation Criteria:**

Yes. The authors focus on synthetic Gaussian mixture scenarios to showcase how BCL escapes local optima, which is a fitting choice for illustrating local-minimum structures in a transparent, controlled manner. In that sense, the method and evaluation align well: they aim to show that combining multiple sub-objectives can overcome entrenched local minima, and Gaussian mixtures are a clean platform to test this claim.

However, because the evaluation remains largely in synthetic domains, real-world applicability (e.g., large-scale neural network tasks) is less directly demonstrated. Nevertheless, for the paper’s stated objective—verifying BCL’s ability to avoid local minima—the chosen GMM-based setup provides appropriate and understandable evidence.

**Other Comments Or Suggestions:**

NA

**Other Strengths And Weaknesses:**

NA

**Questions For Authors:**

NA

**Relation To Broader Scientific Literature:**

Local Optima: Builds on existing work addressing this problem (e.g., simulated annealing) and connects directly to studies on local optima in GMMs.

Learning from Foundation Models: Leverages the success of models like GPT/BERT by analyzing their learning processes, emphasizing the importance of diverse information utilization.

Task Diversity: Relates to multi-task/curriculum learning but focuses on designing tasks with different local optima to actively escape poor solutions. This idea also generalizes the data augmentation and masking approaches of foundation models (like BERT or GPT), which employ specific subsets or transformations of features in training.

In essence, the paper introduces a novel learning paradigm inspired by foundation models to tackle the local optima problem, drawing upon and extending existing concepts in machine learning.

**Theoretical Claims:**

No  theoretical proofs

---

> ### Author Rebuttal · Authors · 2025-04-01
>
> We sincerely appreciate your thoughtful review and recognition of the contributions of our work. Below, we systematically address each of the raised concerns with additional evidence where appropriate. We welcome further feedback.
>
> **Q1: Statistical significance tests**
>
> In the FKL experiments, Fig. 2 (b) has shown the quantitative results over 100 random runs, where BCL stably delivers the global optimum (evident from the near-zero Test Joint FKL values and supported by Fig. 2(a)), while Joint Matching fails to reach the global optimum with probability 1.
>
> To further validate BCL's effectiveness, we have conducted additional experiments:
>
> - We extend the FKL experiments to **8 real-world clustering datasets**, where BCL shows boosted performance over SOTA clustering methods. Details can be found in our responses to Reviewer YmdK’s Q4.
>
> - We repeat the RKL experiments in Fig. 3 over 100 random runs. The results show that Joint Matching almost always gets stuck in the 1-mode local optimum (with a Test Joint RKL of 3.211±0.036), while our BCL effectively conquers this strong local optimum (delivering a Test Joint RKL of **0.362±0.155** and exploring **86.6%** modes on average within 10K iterations; more iterations would improve the performance). It's worth noting that Conditional Matching in the original domain gets stuck in the same local optimum and delivers roughly the same performance as Joint Matching, similar to what is shown in Appendix Fig. 4.
>
> - We perform ablation studies for both FKL and RKL experiments, based on 10 runs. The results are summarized in the following two tables, where JM/MM/CM/RT denote joint/marginal/conditional/randomly-transformed matching, respectively. These studies clearly show that random transformations, which greatly increase the diversity of matching tasks, play a key role in BCL, especially in RKL scenarios.
>
> **Table 1: Ablation study for the FKL experiment**
> |Training Method|Test **Joint FKL**|
> |------|------|
> |JM FKL|0.263±0.035|
> |+MM|0.141±0.054|
> |+MM+CM|0.124±0.044|
> |+MM+RT (BCL)|**0.030+0.006**|
>
> **Table 2: Ablation study for the RKL experiment**
> |Training Method|Test **Joint RKL**|
> |------|------|
> |JM RKL|3.219±0.000|
> |+MM|3.201±0.056|
> |+MM+CM|3.219±0.000|
> |+MM+RT (BCL)|**0.407±0.238**|
> |JM RKL (Adam)|3.219±0.000|
>
> **Q2: Generalization to large-scale neural networks … theoretical evidence proving that BCL always overcomes local optima under broader conditions … it universally conquers local optima**
>
> Developing a unified approach that **uniformly addresses local optima across multiple representative learning paradigms** is both fundamentally valuable and challenging, even in controlled GMM scenarios. Please see our responses to Reviewer YmdK’s Q1.
>
> While BCL represents a promising starting point, generalizing what we’ve done on GMMs to DNN scenarios remains a long-term challenge, because
>
> - in order to conquer the local optima in DNN scenarios, one must first “understand” their local optima, which unfortunately remains largely unexplored, especially considering the uniformity across multiple learning paradigms;
>
> - for Foundation Models (FMs), even defining local/global optima is challenging given their vast training data and diverse downstream applications;
>
> - the empirical research required is computationally intensive; and
>
> - if one proposes a theory that universally conquers local optima under broad conditions (i.e., the theory always finds the global optimum), it revolutionizes machine learning.
>
> We position BCL as a first step towards this ultimate goal. Our transparent research in controlled GMM scenarios is expected to help filter out promising future directions.
>
> **Q3: Strong assumptions about data/model capacity**
>
> As explained in our response to Q2, uniformly addressing the local-optima challenge across multiple paradigms is inherently difficult. In order to clearly demonstrate the potential of BCL in a transparent and easy-to-understand way, we have made these assumptions to isolate it from confounding factors such as non-ideal data and imperfect model capacity. Please also see the additional discussions in our responses to Reviewer w4Bu’s Q1.
>
> In practical applications, approximate fulfilment of these assumptions may be sufficient. In particular:
>
> - The iid data and sufficient model capacity assumptions are well-established conventions in deep learning practice.
>
> - Existing FMs show that one can use a universal DNN to approximate different $p_{\theta}(y_{T}|y_{S})$s, which has led to widespread success in many numerous practical applications. Please also see our responses to Reviewer w4Bu’s Q4.
>
> **Q4: data augmentation**
>
> The ideal data assumption inherently obviates the need for data augmentation; see Footnote 1.

---

### Official Review · Reviewer_w4Bu · 2025-03-12

**Overall Recommendation:** 2

**Summary:**

This paper introduces "big cooperative learning" (BCL), a learning approach to address local optima challenges in conventional machine learning paradigms. The core concept involves diversely exploiting available information (data samples or energy landscapes) to design multiple cooperative training tasks with different local optima but sharing the same global optimum. BCL claims to destabilize local optima by randomly switching among these tasks while encouraging exploration toward the global optimum.
The authors demonstrate BCL using Gaussian Mixture Models (GMMs) in tailored simulations, focusing on both forward KL (FKL) minimization (related to maximum likelihood learning) and reverse KL (RKL) minimization (related to adversarial learning). The paper positions BCL as a generalization of training methods used in foundation models, drawing parallels between BCL's diverse task creation and the varied exploitation of information in models like BERT and GPT.

## update after rebuttal
While the reviewers addressed my concerns, I think my rating is still valid. While I think the paper has potential, I think it would benefit from improvements in writing and defining its story, as well as the directions and shortcomings I outlined below.

**Claims And Evidence:**

While the paper presents interesting ideas, several claims are not adequately supported:

- The claim that BCL is the "missing core element" from conventional learning paradigms is overstated given the limited scope of experiments. The authors present BCL as a revolutionary approach but primarily demonstrate it on controlled GMM simulations.
- The evidence for BCL's effectiveness is primarily qualitative and visual, lacking rigorous quantitative metrics that would strengthen the case. The 25-GMM simulations show promising results but do not include statistical validation across multiple runs or comparison to baseline methods.
- The connection between foundation model training and BCL is plausible but not sufficiently validated. The authors draw parallels but don't demonstrate that the success of foundation models is primarily due to the mechanism they've isolated. For example, an ablation study over the different objectives would greatly strengthen the paper.
The paper claims that BCL delivers the "emerging power of exploration" but doesn't quantitatively measure exploration capabilities compared to established techniques.

**Essential References Not Discussed:**

To be fair, the authors are addressing a fundamental problem of Machine Learning. A comprehensive comparison with related work would be beyond the 8-page limit.
Nonetheless, the current embedding of BCL is rather superficial, mentioning well-known limitations of mode-seeking and mode-covering. I think the paper would benefit from a more in-depth comparison to on-going work in this field.

For addressing local optima:
- Entropy-regularized methods (e.g., Entropy-SGD by Chaudhari et al., 2016)
- Cyclical learning rates (Smith, 2017)

For diverse information exploitation:
- Multi-view learning (Xu et al., 2013)
- Self-supervised contrastive learning (Chen et al., 2020), which also exploits data transformations

For exploration in learning:
- Thompson sampling and other exploration strategies
- Curiosity-driven learning (Pathak et al., 2017)

For mode-seeking/covering behavior:
- More recent GAN variants addressing mode collapse (e.g., VEEGAN is cited, but PacGAN, MSGAN are not)

**Experimental Designs Or Analyses:**

The experimental designs are thoughtfully constructed to demonstrate the core principles of BCL:

- The 2D GMM-based visualizations effectively show how diverse tasks have different local optima landscapes but share the same global optimum.
- The 25-component GMM simulations appropriately test BCL in more challenging scenarios for both FKL and RKL minimization.
- The deliberate initialization of the RKL experiment to encourage mode collapse (placing all components in one corner) provides a strong challenge for testing BCL's exploration capabilities.

However, the experiment designs have several weaknesses:

- The experiments are limited to synthetic data and controlled settings.
- There are no comparisons with other methods designed to address local optima (e.g., simulated annealing, cyclic learning rates, or noise injection).
- The paper lacks ablation studies that would isolate the impact of different components of BCL.

I would have also liked to see more quantitative metrics beyond visual demonstrations, a sensitivity analysis to different hyperparameters of the method and statistical validation of results across multiple runs.

**Methods And Evaluation Criteria:**

The methods are reasonable for a proof-of-concept but have significant limitations:

- The exclusive use of GMMs, while justified for interpretability, raises questions about generalizability to more complex models and real-world problems.
- The evaluation lacks established metrics to quantify improvements in avoiding local optima or exploration efficiency.
- The paper focuses on controlled simulations rather than challenging real-world datasets, limiting the findings' practical impact.
- The authors acknowledge potential challenges in scaling BCL to DNNs (Remark 3.4) but don't adequately address how these would be overcome in practice.

**Other Comments Or Suggestions:**

- The paper would benefit from more rigorous empirical evaluation, including quantitative metrics and statistical validation.
- A more concrete algorithm formulation would help clarify how BCL should be implemented in practice.
- Application to at least one real-world dataset, even with a simplified model, would strengthen the practical relevance.
- A comparison with other methods addressing local optima would provide context for BCL's contributions.
- The computational overhead of task switching and its impact on convergence could be analyzed.
- Maybe the authors presented it and it flew over my head, but it'd be great if they could clarify again how Table 1 and BCL are connected and the role of the transformation function g through an ablation study.

**Other Strengths And Weaknesses:**

Strengths:

- The paper proposes an interesting conceptual framework that connects foundation model training to fundamental learning challenges.
- The visualizations effectively illustrate how different tasks have different local optima landscapes.
- The application to both FKL and RKL minimization demonstrates some versatility.

Weaknesses:

- The paper lacks empirical validation beyond GMMs and controlled settings, raising significant questions about practical applicability.
- The computational feasibility of BCL for large-scale problems is not adequately addressed.
- The paper doesn't provide a clear algorithm or procedure for implementing BCL in practice, particularly for DNNs.
- There's a lack of comparative evaluation against alternative methods for addressing local optima.
- The paper claims BCL as a fundamental missing element but doesn't convincingly demonstrate that it's the key factor in foundation model success, i.e., some kind of ablation study would be greatly appreciated.

**Questions For Authors:**

- How would you implement BCL for DNNs in practice? Remark 3.4 acknowledges challenges in ensuring interrelationships among different $p_{\theta}(y_{T}|y_{S})$, but doesn't provide a concrete solution. Without addressing this, the practical applicability of BCL remains uncertain.
- Have you conducted any experiments comparing BCL with established methods for addressing local optima (e.g., simulated annealing, momentum methods, noise injection)? Such comparisons are essential to validate the claimed advantages of BCL.
- What is the computational overhead of implementing BCL compared to conventional learning? The paper doesn't address how the task switching mechanism affects convergence speed and overall training efficiency.
- How robust is BCL to the choice of task distribution and transformation distribution? Are there optimal strategies for selecting these, or does BCL require extensive tuning?
- The paper claims that BCL is a key element missing from conventional learning paradigms, but many recent advances have occurred without it. What evidence suggests that BCL, rather than other factors like model scale or data quality, is truly the missing core element? What I am most confused about is if the GPT models are so amazing with Next-Token Prediction, why not try to rephrase every problem as a next-token prediction problem. BCL is saying, mix the objectives in a multi-task manner, motivated by escaping local minima, but the connection to the success of LLMs, which use next-token prediction, is vague.

To encourage the authors, I think the work done in the paper is solid and I am intrigued by the idea. I personally haven't considered mixing all these objectives and I am curious about how this could be feasible with deep neural networks. I am keen to change my score and engage in discussion, and a more in-depth ablation over the different objectives would affect my rating greatly.

**Relation To Broader Scientific Literature:**

The paper connects to relevant literature but has some gaps:

- It doesn't sufficiently engage with the rich literature on methods to escape local optima beyond citing a few papers on mode collapse.
The discussion of multi-task learning (section 4) acknowledges differences between classical MTL and BCL but doesn't thoroughly examine other works that use task diversity to improve learning.

**Theoretical Claims:**

The theoretical framework is generally sound but has some limitations:

- The mathematical formulation of BCL in section 3.2 is coherent, but the ideal assumptions (section 2.1) significantly simplify the problem.
- The theoretical justification for why task switching helps escape local optima (Remark 3.3) is intuitive but lacks formal proof or guarantees.
- The paper does not provide theoretical convergence analysis or bounds on the performance improvement that BCL might offer.

I do not think that a theoretical convergence analysis or bounds are necessary for acceptance, but I think the claims made in the paper have to be softened.

---

> ### Author Rebuttal · Authors · 2025-04-01
>
> We appreciate your comprehensive comments with insightful future directions. Below we address your main concerns within the 5000-character limit, with some details consolidated in responses to other reviewers. We welcome further discussion.
>
> **Q1: BCL is the missing core element … is overstated. What evidence?**
>
> We will moderate our claims in the revision.
>
> From an application-agnostic data, modeling, and learning perspective, **complex** foundation models (FMs) **greatly succeed** with **imperfect** data and modeling, while **simple** conventional learning paradigms **suffer badly** even with **perfect** data and modeling. Thus, the missing core element must be associated with learning (see the last paragraph of page 1).
>
> **Q2: How are Table 1 and BCL connected?**
>
> Table 1 shows that many FM objectives are special cases of BCL. For example, BCL recovers the next-token prediction by setting $D=FKL$, $g=identity$, $T=t$, $S=<t$, $\rho=\nu$, and $q’=q$; see the paragraph after Eq. (5). BCL thus encapsulates the common learning essence of existing FMs.
>
> Our transparent GMM studies reveal that BCL’s diverse information exploitation conquers local optima and encourages exploring the global optimum. This revelation may extend to FMs to explain, e.g., the success mechanism of next-token prediction. Note also that diverse conditional matching isn't universally effective; see our responses to Reviewer YmdK’s Q7.
>
> **Q3: quantitative … validation across multiple runs … real-world problems … ablation studies**
>
> Fig. 2b has shown the quantitative results over 100 random runs. We perform additional experiments on 8 real-world clustering datasets, repeat the experiments in Fig. 3 over 100 runs, and perform FKL and RKL ablation studies. Please see our responses to Reviewer 4KZD’s Q1.
>
> **Q4: Implementing BCL for DNNs**
>
> Since BCL contains many FM objectives as special cases (see Q2), its DNN implementation can follow existing FMs.
>
> We expose the issue that the interrelationship (i.e., Bayes’ rule) among $p_{\theta}(y_T|y_S)$s is ignored in existing FMs; this is considered a small contribution. To address the issue, we’d use $p_{\theta}(y_{T}|y_{S})$ to generate pseudo data, which are then used to form additional regularization tasks (similar to Eq. (6)) to promote Bayes' rule compliance.
>
> **Q5: On insightful future research directions for BCL**
>
> Revealing the success mechanism of FMs, either empirically or theoretically, is challenging given their unexplored local optima and broad downstream applications. See our responses to Reviewer 4KZD’s Q2 for details.
>
> Before quantifying the conquest of local/global optima in DNN scenarios, one must first quantify the local/global optima themselves. Both tasks are clearly beyond the scope of this paper. We have shown in Fig. 2 that BCL stably delivers the global optimum in controlled GMM scenarios while Joint Matching always fails. Please see other experiments in our responses to Reviewer 4KZD’s Q1, where we have quantitatively measured exploration.
>
> **Q6: On the ideal assumptions**
>
> Please see our responses to Reviewer 4KZD’s Q3.
>
> **Q7: Comparisons with other methods across many applications**
>
> We position this paper as the first transparent research towards a unified approach that **uniformly addresses local optima across multiple learning paradigms**. See our responses to Reviewer YmdK’s Q1 and the additional experiments in our responses to Reviewer 4KZD’s Q1.
>
> Therefore, extending BCL to various applications (e.g., Bandits/RL) and making application-specific comparisons therein are orthogonal to the scope of this paper. We’ll discuss the related applications and suggested references in the revision.
>
> The way BCL conquers local optima (i.e., exploiting many cooperative objectives) is orthogonal to what existing techniques do (e.g., simulated annealing and momentum are designed for one objective). Extending BCL with these techniques is left to future research. Empirically, Joint Matching with Adam fails with probability 1 in the RKL experiments; see our responses to Reviewer 4KZD’s Q1.
>
> The ideal data assumption eliminates the need for data augmentation; see Footnote 1.
>
> **Q8: Computational overhead of BCL … algorithm formulation**
>
> Definition 3.1 says “one task at a time”. Accordingly, BCL’s computational overhead comes from sampling a task (i.e., $(S, T, g)$) and computing $y=g(x)$; both are lightweight. While BCL may require more iterations, it often produces a much better solution (see Figs. 2-3).
>
> We will add algorithm formulations to the revised appendix.
>
> **Q9: BCL’s robustness to task scope and optimal strategy for selecting tasks**
>
> Our experience is that if the tasks are sufficiently diverse, BCL robustly conquers local optima with minimal tuning; see Remark 3.3. While we used uniform task sampling in all experiments, smart next-task selection (i.e., learning to big learn) could definitely improve exploration, as noted in Concluding Remarks.

---

### Official Review · Reviewer_YmdK · 2025-03-13

**Overall Recommendation:** 2

**Summary:**

This paper focuses on the generative model and discusses several learning objectives, regardless of the model architecture and data.
The authors generalize the conventional learning objective and conditional learning objective to propose the Big Learning, aiming to eliminate the local minima problem.
The authors verify the effectiveness of Big Learning on GMMs.

## update after rebuttal
Thank you for the response. Some of my concerns are addressed. However, given the current presentation (writing, etc), I am still lean to rejection. Nevertheless, I raise my rating to 2.

**Claims And Evidence:**

1. The authors claims that Big Learning is designed to incorporate multi-task learning. However, no experiment is designed to demonstrate the ability of Big Learning in multi-task learning.
2. The Big Learning is seemingly general learning objective which combines both joint matching and conditional matching. Despite its good intuition, there is no evident theoretical support for the superior performance of Big Learning over joint matching and conditional matching.

**Essential References Not Discussed:**

N/A

**Experimental Designs Or Analyses:**

1. The experiments are too toy to demonstrate the effectiveness of Big Learning. The idea of Big Learning originates from the application, such as GAN, GPT, MAE, etc. At least some experiments in more realistic settings are expected to demonstrate the broader application of the proposed method.
2. Furthermore, in Fig. 2 and 3, Big Learning is only compared with Joint Matching. A comparison with Conditional Matching is expected.

**Methods And Evaluation Criteria:**

N/A

**Other Comments Or Suggestions:**

N/A

**Other Strengths And Weaknesses:**

1. In my opinion, the biggest problem of this paper is its writing. It makes the audience to capture the key idea of this paper, especially the introduction part. Actually, the authors introduce the central contribution, e.g., Big Learning, until Page 5, and I finally understand it after reading through the whole paper 3 times.
2. Another major issue is the toy settings in experiments. The authors are suggested to demonstrate the broader application of Big Learning under more realistic settings.
3. Lacks theoretical analysis.

Overall, I am lean to a clear rejection.

**Questions For Authors:**

N/A

**Relation To Broader Scientific Literature:**

N/A

**Theoretical Claims:**

No theoretical contribution.

---

> ### Author Rebuttal · Authors · 2025-03-28
>
> We appreciate your comments. As there are misunderstandings, we invite you to read our responses along with other reviewers’ comments, i.e., the comprehensive and insightful comments of Reviewer w4Bu and the concise and objective assessment of Reviewer 4KZD. We welcome further discussion and thank you for your time.
>
> **Q1: On the value of BCL**
>
> We are addressing the fundamental local-optima challenge of machine learning (noted by Reviewer w4Bu). It’s extremely valuable and challenging to develop a unified approach that **uniformly addresses local optima across multiple learning paradigms**. BCL is the first to demonstrate this potential.
>
> Specifically, BCL simultaneously conquers the entrenched local-optima challenge in FKL (maximum-likelihood) and RKL (adversarial) paradigms by diversely exploiting the available information, as validated in controlled GMM settings (acknowledged by Reviewer 4KZD).
>
>
> **Q2: BCL … combines both joint matching and conditional matching**
>
> Remark 3.2 stated that the BCL can exhaustively cover all joint, marginal, and conditional matching tasks across many transformed domains.
>
> **Q3: On controlled GMM settings**
>
> Following our responses to Q1, to rigorously and transparently study how to uniformly conquer local optima across multiple paradigms, we need a platform that applies to multiple paradigms and has well-studied local optima. DNNs are clearly not an option (see our responses to Reviewers w4Bu's Q5 & 4KZD’s Q2). GMMs provide this clean platform, as recognized by Reviewer 4KZD who noted this as a strength of the paper.
>
> **Q4: Experiments in realistic settings**
>
> We extend the FKL experiments to 8 real-world clustering datasets (https://www.csie.ntu.edu.tw/~cjlin/libsvmtools/datasets/multiclass.html). Compared to SOTA clustering methods such as WM-GMM (A universal framework for learning the elliptical mixture model. TNNLS 2020) and SW-GMM (Sliced Wasserstein distance for learning gaussian mixture models. CVPR 2018), BCL (BigLearn+EM) delivers boosted performance (see results below), justifying its effectiveness. See also other additional experiments in our responses to Reviewer 4KZD’s Q1.
> |Dataset|Metric|WM-GMM|SW-GMM|Joint-EM|BigLearn+EM|
> |------|------|------|------|------|------|
> |Covtype|NMI|0.101±0.0158|0.138±0.0341|0.119±0.0272|**0.182±0.0112**|
> |Covtype|ARI|0.065±0.0497|0.037±0.0658|0.057±0.0216|**0.098±0.0143**|
> |Covtype|Joint-LL|70.957±0.059|70.268±1.632|72.194±1.9089|**74.251±0.8449**|
> |Glass|NMI|0.419±0.0726|0.426±0.0534|0.436±0.0644|**0.450±0.0504**|
> |Glass|ARI|0.198±0.0511|0.195±0.0479|0.220±0.0526|**0.226±0.0434**|
> |Glass|Joint-LL|7.142±0.8023|7.148±0.9546|7.008±1.0364|**7.204±1.0130**|
> |Letter|NMI|0.279±0.0033|0.478±0.0186|0.492±0.0169|**0.544±0.0106**|
> |Letter|ARI|0.012±0.0032|0.190±0.0202|0.193±0.0181|**0.255±0.0163**|
> |Letter|Joint-LL|12.38±0.1024|19.045±0.1548|19.297±0.1877|**19.905±0.0718**|
> |Pendigits|NMI|0.782±0.0233|0.744±0.0385|0.771±0.0323|**0.831±0.0109**|
> |Pendigits|ARI|0.679±0.0487|0.600±0.0663|0.626±0.0622|**0.741±0.0185**|
> |Pendigits|Joint-LL|10.068±0.1824|9.870±0.2198|9.960±0.2545|**10.370±0.0396**|
> |Satimage|NMI|0.575±0.0256|0.598±0.0542|0.587±0.0311|**0.612±0.0195**|
> |Satimage|ARI|0.498±0.0451|0.505±0.1562|0.470±0.0765|**0.520±0.0181**|
> |Satimage|Joint-LL|39.214±0.0035|39.384±0.092|39.387±0.0062|**39.508±0.0437**|
> |Seismic|NMI|0.167±0.0145|0.196±0.0090|0.198±0.0259|**0.224±0.0055**|
> |Seismic|ARI|0.113±0.1584|0.089±0.0426|0.057±0.0292|**0.165±0.0104**|
> |Seismic|Joint-LL|41.958±0.2185|42.234±0.1441|42.050±0.8780|**42.619±0.0367**|
> |Svmguide2|NMI|0.098±0.0372|0.108±0.0638|0.085±0.0746|**0.215±0.0747**|
> |Svmguide2|ARI|0.061±0.0348|0.087±0.0911|0.050±0.0820|**0.225±0.0918**|
> |Svmguide2|Joint-LL|10.248±0.0546|**10.416±0.4158**|10.404±0.4240|10.410±0.3951|
> |Vehicle|NMI|0.218±0.0152|0.178±0.0545|0.197±0.0655|**0.230±0.0330**|
> |Vehicle|ARI|0.102±0.0131|0.085±0.0533|0.094±0.0476|**0.128±0.0281**|
> |Vehicle|Joint-LL|22.300±1.0494|22.473±1.0635|22.896±1.3036|**23.893±1.3625**|
>
> **Q5: BCL vs multitask learning**
>
> As discussed in the first part of Related Work, BCL, which conquers local optima with cooperative tasks, differs fundamentally from conventional multitask learning.
>
> **Q6: Lacks theoretical analysis**
>
> Even for GMMs, SOTA theoretical research is still analyzing their local optima in FKL territory (Chen et al., 2024b), let alone conquering them uniformly across multiple paradigms. See also our responses to Reviewer 4KZD’s Q2.
>
> **Q7: BCL vs Conditional Matching (CM)**
>
> Fig. 1 and Appendix Fig. 4 clearly show BCL’s advantages over CM. Even in this simple setup, combining different CMs (Fig.4) brings no benefits, confirming it's not the unified approach in our responses to Q1.
>
> Since BCL doesn't use CM in Fig. 2 (see Lines 401-404, left), we compare BCL with CM in the RKL experiments in Fig. 3. The results (in our responses to Reviewer 4KZD’s Q1) show that CM gets stuck in the same local optimum as Joint Matching.

---

### Decision · Program_Chairs · 2025-05-01

**Decision:**

Reject

**Comment:**

The paper proposes "big cooperative learning" to overcome local optima in multimodal energy landscapes. The approach is based on random local projections to optimize the global landscape. The authors introduce a learning objective and related it to prior work based on forward and reverse KL optimization. They claim the approach is general. Demonstrations are on mixture of gaussian problems. There is no theory.

Multiple reviewers note that the writing could be significantly improved. I agree. There are several challenges in writing that obfuscate the core claims, including the name, which is insufficiently descriptive of the approach and potentially confusing. Reviewers also note the lack of theory. Some kind of theoretical analysis is important if one wants to claim a general optimization approach. The limited nature of the data was also noted by reviewers. Given the grand claims, it would be important to consider datasets that are more in line with state of the art. A systematic comparison of other methods, especially ones that highlight why the approach works, would be greatly appreciated. For example, a classic would be Rasmussen's (2000) Infinite mixture of gaussians model using gibbs sampling, and perhaps subsequent variational approximations.